# Towards Faithful XAI Evaluation via Generalization-Limited Backdoor Watermark

**Mengxi Ya**[1*]**, Yiming Li**[2,3,*,†]**, Tao Dai**[4]**, Bin Wang**[1,5]**, Yong Jiang**[1]**, Shu-Tao Xia**[1]

[1]Tsinghua Shenzhen International Graduate School, Tsinghua University, Shenzhen, China
[2]The State Key Laboratory of Blockchain and Data Security, Zhejiang University, China
[3]ZJU-Hangzhou Global Scientific and Technological Innovation Center, China
[4]College of Computer Science and Software Engineering, Shenzhen University, China
[5]Guangzhou Intelligence Communications Technology Co.,Ltd., China
{yamx21, w-b21}@mails.tsinghua.edu.cn; li-ym@zju.edu.cn; daitao.edu@gmail.com;
{jiangy, xiast}@sz.tsinghua.edu.cn;

## Abstract

Saliency-based representation visualization (SRV) (*e.g.*, Grad-CAM) is one of the most classical and widely adopted explainable artificial intelligence (XAI) methods for its simplicity and efficiency. It can be used to interpret deep neural networks by locating saliency areas contributing the most to their predictions. However, it is difficult to automatically measure and evaluate the performance of SRV methods due to the lack of ground-truth salience areas of samples. In this paper, we revisit the backdoor-based SRV evaluation, which is currently the only feasible method to alleviate the previous problem. We first reveal its *implementation limitations* and *unreliable nature* due to the trigger generalization of existing backdoor watermarks. Given these findings, we propose a generalization-limited backdoor watermark (GLBW), based on which we design a more faithful XAI evaluation. Specifically, we formulate the training of watermarked DNNs as a min-max problem, where we find the 'worst' potential trigger (with the highest attack effectiveness and differences from the ground-truth trigger) via inner maximization and minimize its effects and the loss over benign and poisoned samples via outer minimization in each iteration. In particular, we design an adaptive optimization method to find desired potential triggers in each inner maximization. Extensive experiments on benchmark datasets are conducted, verifying the effectiveness of our generalization-limited watermark. Our codes are available at `https://github.com/yamengxi/GLBW`.

## 1 Introduction

Deep neural networks (DNNs) have been widely and successfully adopted in many areas, such as facial recognition (Tang & Li, 2004; Qiu et al., 2021; Xia et al., 2023). Despite their promising performance, DNNs are also criticized for acting as a 'black box' whose predictions are not interpretable. The lack of interpretability of DNNs is mostly due to their nonlinear components (*e.g.*, activation functions) and multi-layer structure, hindering DNNs to be deployed in mission-critical applications (*e.g.*, medical diagnosis) at scale.

Currently, there are tremendous efforts to explain their prediction behaviors and mechanisms. These methods are called explainable artificial intelligence (XAI). Current, existing XAI methods can be roughly divided into three main categories, including **(1)** visualizing DNN representations (Zeiler & Fergus, 2014; Wang et al., 2020; Rao et al., 2022), **(2)** distilling DNNs to explainable models (Li et al., 2020b;a; Ha et al., 2021), and **(3)** building explainable DNNs (Zhang et al., 2018; Xiang et al., 2020; Li et al., 2022a). Among all these methods, saliency-based representation visualization (SRV) (Springenberg et al., 2015; Selvaraju et al., 2017), as a sub-category of the first one, is probably the most classical and widely adopted one for its simplicity and efficiency. In general, these methods try to measure the significance of each input region to the final prediction. Accordingly, they can locate and highlight saliency input areas that contribute the most to a model prediction.

---

[*]The first two authors contributed equally to this work. Correspondence to: Yiming Li (li-ym@zju.edu.cn).

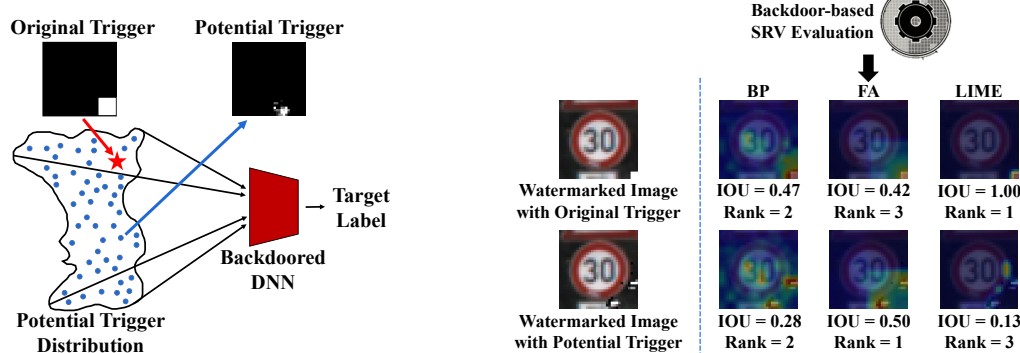

Figure 1: The generalization of existing backdoor watermarks where all potential triggers can activate backdoors.

Figure 2: The backdoor-based evaluation of three SRV methods (*i.e.*, BP, FA, and LIME) with original and potential triggers.

However, it is difficult to measure and evaluate the performance of SRV methods due to the lack of the ground-truth salience map of the sample. Currently, the most common evaluation method is to have human experts score or rank saliency maps of different SRV methods for each sample. This method is time-consuming, expensive, and may be subject to significant deviations due to the bias of different people. To alleviate this problem, (Lin et al., 2021) proposed to exploit (patch-based) backdoor attacks (Li et al., 2022c) (*e.g.*, BadNets (Gu et al., 2019)) to design an automatic and efficient XAI evaluation. Specifically, they modified a few training samples by stamping a pre-defined trigger patch on their images and changing their labels to a specific target label. DNNs trained on the modified dataset will learn a latent connection between the trigger patch and the target label (*i.e.*, the backdoor). Accordingly, all samples containing the trigger patch will be predicted to be the target label, no matter what their ground-truth label is. Given this backdoor property, they can calculate the average intersection over union (IOU) between the saliency areas generated by the SRV method of the backdoored model over different backdoored samples and the trigger area as an indicator to evaluate the SRV method. This method is developed based on the understanding that trigger regions can be regarded as areas that contribute the most to the model's predictions and therefore can serve as the ground-truth salience areas for evaluation.

In this paper, we revisit the backdoor-based SRV evaluation. We first reveal that it has three implementation limitations, including **(1)** failing to take absolute values of gradients in the calculation of some SRV methods, **(2)** selecting saliency areas based on the threshold instead of the saliency size, and **(3)** using the minimum bounding box covering all edges of significant regions instead of the regions themselves for calculating IOU of SRV methods, leading to potentially false results. More importantly, this method relies on a latent assumption that *existing backdoor attacks have no trigger generalization*, *i.e.*, only the trigger used for training (dubbed 'original trigger') can activate backdoors, since they treat trigger regions as the ground-truth salience areas. We show that *this assumption does not hold for existing backdoor attacks and therefore may lead to unreliable results*. For example, given a SRV method, assume that its generated saliency areas of most poisoned samples are only a small part of that of the original trigger. According to the existing backdoor-based evaluation, this SRV method will be treated as having poor performance since it has a small IOU. However, due to the generalization of backdoor watermarks, the model may only learn this local region rather than the whole trigger. In this case, the evaluated SRV method is in fact highly effective, contradicting to the results of the backdoor-based SRV evaluation. Besides, existing backdoor-based evaluation has different results with potential trigger patterns that can activate the backdoor while significantly different from the original trigger (as shown in Figure 1-2). This phenomenon further verifying that its results are unreliable. Accordingly, an intriguing and important question arises: *Is it possible to design a faithful automatic XAI evaluation method*?

In this paper, we explore how to reduce the position generalization of backdoor triggers, based on which to design a faithful backdoor-based XAI evaluation. Arguably, the most straightforward method is to synthesize the potential trigger pattern and penalize its effects based on its distance to the original trigger in each iteration during the training process. However, as we will show in our main experiments, this method suffers from relatively poor performance (*i.e.*, still with high trigger generalization) in many cases. This failure is mostly because the synthesized triggers are 'weak' with either a large loss value or are similar to the original trigger. Based on these understandings, we formulate the training of watermarked DNNs as a min-max problem, where we find the 'worst' potential trigger with the highest attack effectiveness and differences from the original trigger via inner maximization and minimize its effects and the loss over benign and poisoned samples through outer

minimization. In particular, we design an adaptive optimization method to find desired ('worst') potential triggers in each inner maximization.

In conclusion, the main contributions of this paper are four-folds: **(1)** We reveal and address the implementation limitations of the existing backdoor-based XAI evaluation; **(2)** We analyze the generalization pattern of backdoor triggers, based on which we further reveal the unreliable nature of the existing backdoor-based XAI evaluation; **(3)** We propose a generalization-limited backdoor watermark (GLBW), based on which we design a more faithful XAI evaluation. To the best of our knowledge, we are the first trying to measure and even manipulate trigger generalization; **(4)** We conduct extensive experiments on benchmark datasets to verify the effectiveness of our GLBW.

## 2 BACKGROUND AND RELATED WORK

### 2.1 BACKDOOR ATTACKS AND THEIR POSITIVE APPLICATIONS

Backdoor attack is an emerging threat to deep neural networks (Li et al., 2022c). Currently, existing backdoor attacks can be roughly divided into two main categories, including patch-based attacks and non-patch-based attacks, based on the trigger properties.

Specifically, patch-based attacks (Gu et al., 2019; Li et al., 2022e; Gao et al., 2023b) adopted some local image patches as the trigger patterns. For example, BadNets randomly modified a few samples from the original benign dataset by stamping the black-and-white trigger square to their images and changing their labels to the target label. These attacks are highly effective since their trigger patterns are significantly different from the surrounding of the replaced image areas in the poisoned samples; In contrast, non-patch-based attacks exploited some transformations over the whole image, such as additive noises (Li et al., 2021b), image-warping (Nguyen & Tran, 2021), and color shifting (Gong et al., 2023), to generate poisoned images. In particular, a few works (Qiao et al., 2019; Li et al., 2021a; Qi et al., 2023) initially discovered trigger generalization phenomenon, $i.e.$, some potential trigger patterns are (significantly) different from the one used for training but can still activate hidden backdoors in the attacked models, in a few cases of neural cleanse (Wang et al., 2019). However, none of them tried to provided its statistical patterns and measure or control this generalization. How to better understand and control trigger generalization is still an important open-question.

Except for using for malicious purposes, there are also a few research studied how to exploit backdoor attacks for positive purposes based on their properties, such as adversarial defense (Shan et al., 2020) and copyright protection (Li et al., 2022b; 2023b; Guo et al., 2023; Gan et al., 2023). In particular, Lin *et al.* (Lin et al., 2021) used backdoor attacks to watermark DNNs, based on which to design an evaluation method of saliency-based representation visualization. However, almost all these applications relied on a latent assumption that backdoor triggers have mild generalization, which does not holds for existing backdoor attacks.

### 2.2 SALIENCY-BASED REPRESENTATION VISUALIZATION (SRV) AND ITS EVALUATIONS

How to explain the predictions or mechanisms of DNNs is always an important research direction. Currently, existing XAI methods of DNNs can be divided into three main categories, including **(1)** visualizing DNN representations (Zeiler & Fergus, 2014; Rao et al., 2022; Chen et al., 2024), **(2)** distilling DNNs to explainable models (Li et al., 2020a; Ha et al., 2021), and **(3)** building explainable DNNs (Zhang et al., 2018; Xiang et al., 2020; Li et al., 2022a). In this paper, we focus on the saliency-based representation visualization (SRV), which a sub-type of the first category.

**Saliency-based Representation Visualization (SRV).** In general, SRV methods explore visual patterns within a DNN unit ($e.g.$, the logit). Specifically, they generate a *saliency map* for each sample whose value indicates the contribution of its corresponding part ($e.g.$, pixel) to the unit's result. In general, we can divide existing SRV methods into two main categories, including white-box SRV and black-box SRV. Their difference lies in whether the SRV method can access the source files of the model. Currently, gradient-based methods are the mainstay of white-box SRV. For example, back-propagation (BP) adopted the gradient of samples concerning their predicted results to render their saliency maps; Guided back-propagation (GBP) (Springenberg et al., 2015) only considered the positive gradients and set the negative parts as zero in ReLU layers; Gradient-weighted class activation mapping (GCAM) (Selvaraju et al., 2017) generated saliency maps via features in the

last convolutional layer; GGCAM (Selvaraju et al., 2017) combined GBP and GCAM via element-wise multiplication. Black-box SRV usually obtained the saliency map by perturbing the input and observing prediction changes. For example, occlusion sensitivity (OCC) (Zeiler & Fergus, 2014) occluded different image portions and monitored output differences; LIME (Ribeiro et al., 2016) learned an interpretable model locally varound the prediction; Recently, feature ablation (FA) (Meyes et al., 2019) perturbed a group of features instead of a single one to render saliency maps.

**The Evaluation Methods of SRV Methods.** In general, it is difficult to measure and evaluate the performance of SRV methods due to the lack of the ground-truth salience map of a sample. Currently, the most reliable evaluation of SRV methods is still based on the human inspection with ranking or scoring (Ribeiro et al., 2016; Kim et al., 2018; Dwivedi et al., 2023). However, this evaluation is time-consuming and costly or even leads to biased and inaccurate results (Buçinca et al., 2020). There are also a few automatic SRV evaluations, mostly by measuring the performance fluctuations when perturbing critical regions (Samek et al., 2016; Yeh et al., 2019; van der Waa et al., 2021). However, these methods usually require heavy computations and are even inaccurate due to distribution shift (Hooker et al., 2019); Recently, Lin *et al.* (Lin et al., 2021) proposed to exploit backdoor-based model watermarks to design SRV evaluation, based on the principle that trigger locations can be regarded as the most critical regions contributing the most to model's predictions. Specifically, it first generated the saliency maps of some poisoned samples containing trigger patterns and turned them into 0-1 masks based on pre-defined thresholds. After that, it found a minimum bounding box covering all edges of the trigger mask detected by the Canny algorithm (Canny, 1986). The final score was calculated by the distance, such as average intersection over union (IOU), between generated bounding box and the trigger mask over all samples. However, as we will show in the following sections, this evaluation method is also unreliable.

## 3 REVISITING BACKDOOR-BASED XAI EVALUATION

### 3.1 STANDARDIZE BACKDOOR-BASED XAI EVALUATION

Currently, the backdoor-based SRV evaluation (Lin et al., 2021) still has some implementation limitations that may lead to inaccurate results. In this section, we will standardize it.

**Implementation Limitations and Their Standardization.** Firstly, (Lin et al., 2021) took absolute values of gradients in only parts of (instead of all) evaluated SRV methods before generating their saliency map. Since both positive or negative influences are all important, to ensure a fair comparison, we propose to *conduct it on the gradients of all SRV methods*; Secondly, (Lin et al., 2021) adopted a pre-defined threshold to filter out the most critical regions (*i.e.*, the 0-1 mask) from the saliency map for SRV methods. However, the same threshold value has different influences across samples even for the same SRV method. To ensure a fair comparison, we propose to *select $M$ pixel*

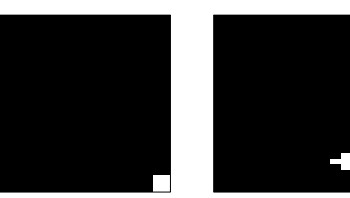

Figure 3: The trigger patterns used for standardized evaluation.

*locations with the maximum saliency value as significant regions*, where $M$ is the size of trigger used for model watermarking; Thirdly, (Lin et al., 2021) adopted the minimum bounding box covering all edges of significant regions instead of themselves for calculating IOU. It may lead to inaccurate results, especially when the original trigger has complex shapes. Accordingly, we *calculate the IOU based on the significant regions directly*. In general, the second limitation has largest effects.

We conduct the following experiments to verify that the vanilla backdoor-based SRV evaluation may lead to inaccurate results and therefore we need to standardize it. Please refer to Appendix C for more details about why Lin et al. (2021) and we only consider patch-based triggers.

**Settings.** We adopt ResNet-18 (He et al., 2016) on CIFAR-10 (Krizhevsky, 2009) and GTSRB (Stallkamp et al., 2012) datasets for our discussions. Following the settings in (Lin et al., 2021), we adopt BadNets (Gu et al., 2019) for model watermarking. Specifically, the target label is set to '0' and the watermarking rate is set to 5%. We adopt four triggers with two patterns (as shown in Figure 3) located in the lower-right corner or a random position of the image. We randomly sample 100 testing images for evaluation. We calculate the average rank by averaging the rank of SRV methods (based on the IOU) in each scenario. Please find more details in Appendix A.

Table 1: The evaluation (IOU) of SRV methods with vanilla backdoor-based method and its standardized version on the CIFAR-10 dataset and the GTSRB dataset. **(a)&(c)**: triggers with the square-like pattern; **(b)&(d)**: triggers with the compass-like pattern; **(a)&(b)**: triggers located in the lower-right corner; **(c)&(d)**: triggers with random locations.

| Method↓ | Dataset↓ | SRV→ Trigger↓ | BP | GBP | GCAM | GGCAM | OCC | FA | LIME |
|---|---|---|---|---|---|---|---|---|---|
| Vanilla | CIFAR-10 | (a) | 0.306 | 0.434 | 0.011 | 0.329 | 0.437 | 0.543 | 0.002 |
| | | (b) | 0.075 | 0.131 | 0.034 | 0.174 | 0.314 | 0.137 | 0.018 |
| | | (c) | 0.248 | 0.344 | 0.014 | 0.308 | 0.386 | 0.394 | 0.006 |
| | | (d) | 0.065 | 0.271 | 0.058 | 0.221 | 0.416 | 0.164 | 0.184 |
| | GTSRB | (a) | 0.519 | 0.428 | 0.000 | 0.373 | 0.466 | 0.640 | 0.006 |
| | | (b) | 0.190 | 0.154 | 0.043 | 0.175 | 0.505 | 0.263 | 0.063 |
| | | (c) | 0.328 | 0.341 | 0.061 | 0.340 | 0.306 | 0.657 | 0.010 |
| | | (d) | 0.233 | 0.207 | 0.102 | 0.210 | 0.549 | 0.198 | 0.087 |
| Standardized | CIFAR-10 | (a) | 0.212 | 0.212 | 0.000 | 0.207 | 0.885 | 0.494 | 0.979 |
| | | (b) | 0.266 | 0.266 | 0.001 | 0.268 | 0.537 | 0.385 | 0.991 |
| | | (c) | 0.346 | 0.346 | 0.000 | 0.348 | 0.928 | 0.579 | 0.964 |
| | | (d) | 0.243 | 0.243 | 0.005 | 0.247 | 0.703 | 0.566 | 0.988 |
| | GTSRB | (a) | 0.344 | 0.344 | 0.000 | 0.315 | 0.739 | 0.423 | 1.000 |
| | | (b) | 0.174 | 0.174 | 0.000 | 0.141 | 0.635 | 0.708 | 0.957 |
| | | (c) | 0.240 | 0.240 | 0.000 | 0.254 | 1.000 | 0.723 | 0.909 |
| | | (d) | 0.087 | 0.087 | 0.012 | 0.090 | 0.721 | 0.608 | 0.879 |

Table 2: The average rank (based on the IOU) of SRV methods that is evaluated with the vanilla backdoor-based method and its standardized version on CIFAR-10 and GTSRB datasets.

| Method↓ | Dataset↓, SRV→ | BP | GBP | GCAM | GGCAM | OCC | FA | LIME |
|---|---|---|---|---|---|---|---|---|
| Vanilla | CIFAR-10 | 5.25 | 3.00 | 6.25 | 3.25 | 1.50 | 2.50 | 6.25 |
| | GTSRB | 2.75 | 3.75 | 6.50 | 3.75 | 2.50 | 2.25 | 6.50 |
| Standardized | CIFAR-10 | 4.75 | 4.75 | 7.00 | 4.50 | 2.00 | 3.00 | 1.00 |
| | GTSRB | 4.50 | 4.50 | 7.00 | 5.00 | 2.00 | 2.75 | 1.25 |

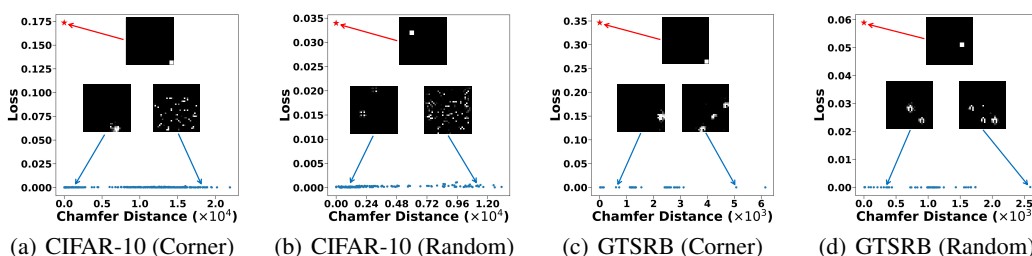

(a) CIFAR-10 (Corner)   (b) CIFAR-10 (Random)   (c) GTSRB (Corner)   (d) GTSRB (Random)

Figure 4: The distance between potential triggers and the original one used for training $w.r.t.$ the loss value on CIFAR-10 and GTSRB. The original trigger is a white square located at the lower-right corner (dubbed 'Corner') or a random position (dubbed 'Random'), denoted by the 'red star'.

**Results.** As shown in Table 1-2, the vanilla backdoor-based method and our standardized version have significantly different results. The vanilla method has various evaluations of the performance of the same SRV method on different datasets (especially for BP). In contrast, our standardized evaluation method has consistent rankings. These results verify that our evaluation is more faithful.

## 3.2 TRIGGER GENERALIZATION AND ITS INFLUENCES TO XAI EVALUATION

**A Closer Look to Location Generalization of Trigger Patterns.** In this section, we generate 1,000 targeted universal adversarial perturbations (with random initializations) regarding the target label of models watermarked with the square-type trigger as potential trigger patterns. The watermarked DNNs are the same as those obtained in Section 3.1. To analyze the generalization properties of backdoor triggers, we visualize the distance between the potential trigger and the ground-truth one $w.r.t.$ its loss value. Specifically, we exploit Chamfer distance (Borgefors, 1988) to cover the locations of all trigger pixels. As shown in Figure 4, many potential trigger patterns are having a large distance to the original one but with significantly lower loss values. In other words, *the trigger pattern of existing watermarks has a high-level position generalization*, although only the original trigger pattern is adopted. It is most probably due to the high non-convexity and excessive learning capacities of DNNs. We will further explore its intrinsic mechanism in our future works.

**Backdoor-based XAI Evaluation with Potential Trigger Patterns.** In this part, we evaluate whether the existing backdoor-based evaluation method has consistent rankings with potential triggers different from the original one but can still activate model backdoors. Specifically, we adopted

Table 3: The IOU evaluated by the standardized backdoor-based method with synthesized triggers.

| Dataset↓ | Trigger↓, SRV→ | BP | GBP | GCAM | GGCAM | OCC | FA | LIME |
|---|---|---|---|---|---|---|---|---|
| CIFAR-10 | Pattern (a) | 0.144 | 0.144 | 0.000 | 0.079 | 0.545 | 0.554 | 0.198 |
| | Pattern (b) | 0.128 | 0.128 | 0.000 | 0.014 | 0.346 | 0.285 | 0.310 |
| | Pattern (c) | 0.007 | 0.007 | 0.037 | 0.008 | 0.064 | 0.065 | 0.070 |
| GTSRB | Pattern (d) | 0.195 | 0.195 | 0.000 | 0.130 | 0.691 | 0.702 | 0.315 |
| | Pattern (e) | 0.224 | 0.224 | 0.000 | 0.107 | 0.562 | 0.573 | 0.342 |
| | Pattern (f) | 0.198 | 0.198 | 0.000 | 0.126 | 0.601 | 0.447 | 0.310 |

Table 4: The average rank (based on the IOU) of SRV methods that is evaluated by the standardized backdoor-based method with different triggers on the CIFAR-10 and the GTSRB datasets.

| Dataset↓ | Trigger↓, SRV→ | BP | GBP | GCAM | GGCAM | OCC | FA | LIME |
|---|---|---|---|---|---|---|---|---|
| CIFAR-10 | Original | 4.75 | 4.75 | 7.00 | 4.50 | 2.00 | 3.00 | 1.00 |
| | Synthesized | 4.67 | 4.67 | 6.00 | 5.67 | 2.00 | 2.00 | 2.00 |
| GTSRB | Original | 4.50 | 4.50 | 7.00 | 5.00 | 2.00 | 2.75 | 1.25 |
| | Synthesized | 4.00 | 4.00 | 7.00 | 6.00 | 1.67 | 1.33 | 3.00 |

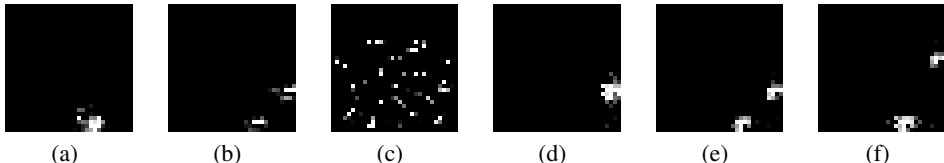

| (a) | (b) | (c) | (d) | (e) | (f) |

Figure 5: The trigger patterns used for generalization discussions.

three synthesized triggers generated in the previous part on each dataset (as shown in Figure 5) and calculate the IOU of SRV methods with them. As shown in Table 3-4, the backdoor-based method has significantly different results across different triggers even on the same dataset (especially for GGCAM, FA and LIME). This phenomenon also indicates that its results are not reliable.

## 4 THE PROPOSED METHOD

As demonstrated in Section 3.2, existing backdoor-based SRV evaluation may have unreliable results due to the trigger generalization of existing backdoor attacks. In this section, we discuss how to obtain the generalization-limited backdoor watermark to design a more faithful SRV evaluation.

### 4.1 PRELIMINARIES

**Threat Model.** In this paper, we assume that users (*i.e.*, SRV appraisers) can fully control the training process of DNNs. They will try to implant backdoor triggers into the model, based on which to evaluate the performance of SRV methods via poisoned samples.

**The Main Pipeline of Backdoor-based Model Watermark.** Let $\mathcal{D} = \{(\boldsymbol{x}_i, y_i)\}_{i=1}^{N}$ denotes the benign training set, where $\boldsymbol{x}_i \in \mathcal{X} = \{0, 1, \dots, 255\}^{C \times H \times W}$ is the image, $y_i \in \mathcal{Y} = \{0, \dots, K-1\}$ is its label, and $K$ is the number of classes. How to generate the watermarked dataset $\mathcal{D}_w$ is the cornerstone of the backdoor-based model watermark. Specifically, $\mathcal{D}_w$ consists of two disjoint parts, including the modified version of a selected subset (*i.e.*, $\mathcal{D}_s$) of $\mathcal{D}$ and remaining benign samples, *i.e.*, $\mathcal{D}_w = \mathcal{D}_m \cup \mathcal{D}_b$, where $y_t$ is a pre-defined target label, $\mathcal{D}_b = \mathcal{D} \backslash \mathcal{D}_s$, $\mathcal{D}_m = \{(\boldsymbol{x}', y_t) | \boldsymbol{x}' = G(\boldsymbol{x}; \boldsymbol{w}), (\boldsymbol{x}, y) \in \mathcal{D}_s\}$, $\gamma \triangleq \frac{|\mathcal{D}_s|}{|\mathcal{D}|}$ is the *watermarking rate*, and $G : \mathcal{X} \rightarrow \mathcal{X}$ is the poisoned image generator with parameter $\boldsymbol{w}$. For example, $G(\boldsymbol{x}) = (\boldsymbol{1} - \boldsymbol{m}) \otimes \boldsymbol{x} + \boldsymbol{m} \otimes \boldsymbol{t}$, where the mask $\boldsymbol{m} \in [0, 1]^{C \times H \times W}$, $\boldsymbol{t} \in \mathcal{X}$ is the trigger pattern, and $\otimes$ is the element-wise product used in the backdoor-based SRV evaluation (Lin et al., 2021). Once the watermarked dataset $\mathcal{D}_w$ is generated, users will train the watermarked DNN $f$ via $\min_{\boldsymbol{\theta}} \sum_{(\boldsymbol{x}, y) \in \mathcal{D}_w} \mathcal{L}(f(\boldsymbol{x}; \boldsymbol{\theta}), y)$.

### 4.2 AN INEFFECTIVE BASELINE: BACKDOOR WATERMARK WITH TRIGGER PENALTY

Arguably, the most straightforward method is to synthesize the potential trigger pattern and penalize its effects based on its distance to the original one in each iteration during the training process. In general, if the potential trigger pattern is close to the ground-truth one, the predictions of watermarked DNNs to samples containing this pattern should be similar to the target label; Otherwise,

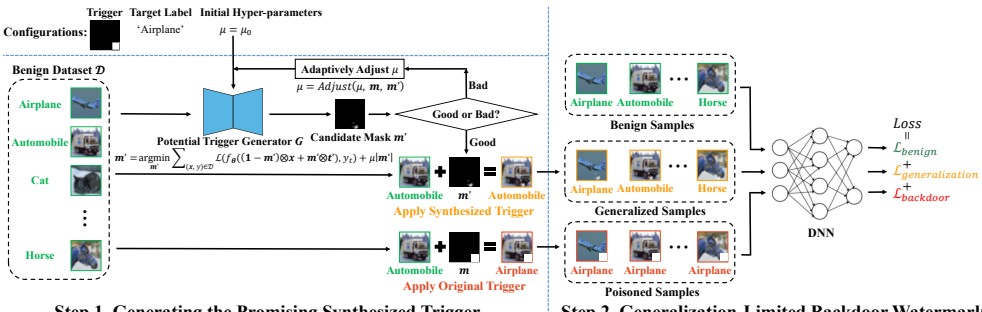

**Step 1. Generating the Promising Synthesized Trigger** | **Step 2. Generalization-Limited Backdoor Watermark**

Figure 6: The main pipeline of our generalization-limited backdoor watermark (GLBW). In each iteration, we first generate promising synthesized patterns with the highest attack effectiveness and differences from the original trigger via adaptively adjusting the trade-off parameter $\mu$; After that, we update the DNN through the weighted combination of benign loss, backdoor loss, and generalization loss. We repeat these two steps until the generalization-limited backdoor watermark is implanted.

their predictions should be similar to their ground-truth label. To achieve this goal, we design a penalty loss and call this method **b**ackdoor **w**atermark with **t**rigger **p**enalty ( **BWTP**), as follows.

$$
\min_{\boldsymbol{\theta}} \sum_{(\boldsymbol{x},y)\in\mathcal{D}} \underbrace{\mathcal{L}\left(f_{\boldsymbol{\theta}}\left(\boldsymbol{x}\right),y\right)}_{Benign\ Loss} + \lambda_1 \cdot \underbrace{\mathcal{L}\left(f_{\boldsymbol{\theta}}\left(\left(\mathbf{1}-\boldsymbol{m}\right)\otimes\boldsymbol{x}+\boldsymbol{m}\otimes\boldsymbol{t}\right),y_t\right)}_{Backdoor\ Loss} +
$$
$$
\lambda_2 \cdot \underbrace{\max_{\boldsymbol{m}',\boldsymbol{t}',|\boldsymbol{m}'|\leq T} \mathcal{L}'\left(f_{\boldsymbol{\theta}}\left(\left(\mathbf{1}-\boldsymbol{m}'\right)\otimes\boldsymbol{x}+\boldsymbol{m}'\otimes\boldsymbol{t}'\right),\hat{\boldsymbol{y}}_t+\hat{\boldsymbol{y}}\right),}_{Penalty\ Loss}
$$
(1)

where $\mathcal{D}$ is the benign dataset, $\mathcal{L}$ and $\mathcal{L}'$ are the loss functions (*e.g.*, cross-entropy and mutual information), $\boldsymbol{t}$ is the original trigger pattern with mask $\boldsymbol{m}$, and $T$ is the threshold of the size of potential trigger mask $\boldsymbol{m}'$. $\hat{\boldsymbol{y}}_t = (1 - p(d(\boldsymbol{m},\boldsymbol{m}'))) \cdot \boldsymbol{y}_t$ and $\hat{\boldsymbol{y}} = p(d(\boldsymbol{m},\boldsymbol{m}')) \cdot \boldsymbol{y}$, where $\boldsymbol{y}$ is the one-hot representation of $y$, $d(\cdot,\cdot)$ is a distance metric (*e.g.*, chamfer distance), $p(\cdot)$ is a probability activation function to map the distance to a probability (the larger the distance, the higher the probability). $\lambda_1$ and $\lambda_2$ are two trade-off hyper-parameters for backdoor loss and penalty loss.

Similar to the optimization strategies used in adversarial training (Madry et al., 2018; Zhang et al., 2019; Li et al., 2022d), we alternately optimize the inner maximization and the outer minimization in each iteration. Please find more optimization details in Appendix D.

### 4.3 GENERALIZATION-LIMITED BACKDOOR WATERMARK (GLBW)

As we will show in the experiments and appendix, the previous BWTP suffers from relatively poor performance whose watermarked DNNs still have high trigger generalization in many cases. According to the intermediate results, this failure is mostly because the synthesized triggers are 'weak' with either a large loss value or are similar to the original trigger. Besides, the training of watermarked DNNs is unstable even when the synthesized triggers are promising. We argue that these limitations are due to the complexity of its penalty loss. As such, we propose to *directly identify the potential trigger patterns that are more different from the ground-truth one and force them to be predicted as the ground-truth label* to further simplify the loss term, inspired by neural cleanse (Wang et al., 2019). Its optimization process is as follows.

$$
\min_{\boldsymbol{\theta}} \sum_{(\boldsymbol{x},y)\in\mathcal{D}} \underbrace{\mathcal{L}\left(f_{\boldsymbol{\theta}}\left(\boldsymbol{x}\right),y\right)}_{Benign\ Loss} + \lambda_3 \cdot \underbrace{\mathcal{L}\left(f_{\boldsymbol{\theta}}\left(\left(\mathbf{1}-\boldsymbol{m}\right)\otimes\boldsymbol{x}+\boldsymbol{m}\otimes\boldsymbol{t}\right),y_t\right)}_{Backdoor\ Loss} +
$$
$$
\lambda_4 \cdot \underbrace{\max_{\boldsymbol{m}',\boldsymbol{t}',|\boldsymbol{m}'\cap\boldsymbol{m}|\leq\tau\cdot|\boldsymbol{m}|} -\left\{\mathcal{L}\left(f_{\boldsymbol{\theta}}\left(\left(\mathbf{1}-\boldsymbol{m}'\right)\otimes\boldsymbol{x}+\boldsymbol{m}'\otimes\boldsymbol{t}'\right),y_t\right)+\mu\cdot|\boldsymbol{m}'|\right\}}_{Generalization\ Loss}
$$
(2)

where $\mu$ is a hyper-parameter trading off its size and effectiveness for synthesized trigger, $\lambda_3$ & $\lambda_4$ are two hyper-parameters for losses, $\tau$ is the overlap threshold between synthesized and original triggers, and $|\boldsymbol{m}'|$ is the size of potential trigger mask. Notice that the $y$ and $y_t$ contained in GLBW are different from $\hat{\boldsymbol{y}}$ and $\hat{\boldsymbol{y}}_t$ used in the previous BWTP since they are fundamentally different.

Table 5: The effectiveness and generalization of model watermarks on CIFAR-10 and GTSRB. '–' denotes that the trigger synthesis method can not find any potential trigger in this case.

| Dataset↓ | Trigger Synthesis Method→ | | | Neural Cleanse | | TABOR | | Pixel Backdoor | |
|---|---|---|---|---|---|---|---|---|---|
| | Metric→
Watermark↓ | BA | WSR | Chamfer | PLG | Chamfer | PLG | Chamfer | PLG |
| CIFAR-10 | Vanilla | 91.73 | 97.39 | 3979.33 | 42.20 | 4030.52 | 18.80 | 1974.68 | 10.30 |
| | BWTP | 85.51 | 89.16 | 8070.46 | 0 | 8615.59 | 0.00 | – | – |
| | GLBW | 78.33 | 93.68 | 23.04 | 100 | 27.92 | 83.33 | 30.16 | 100 |
| GTSRB | Vanilla | 97.57 | 94.52 | 217.71 | 87.20 | 299.05 | 83.39 | 257.15 | 89.50 |
| | BWTP | 88.36 | 91.61 | 21.83 | 100 | 20.16 | 100 | 25.71 | 100 |
| | GLBW | 93.89 | 90.17 | 23.73 | 100 | 34.08 | 85.00 | 33.13 | 100 |

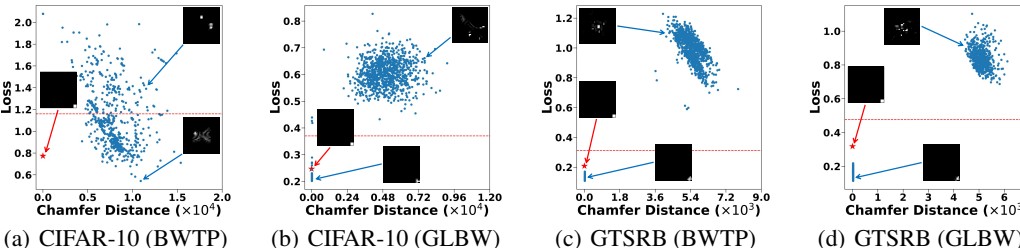

(a) CIFAR-10 (BWTP)   (b) CIFAR-10 (GLBW)   (c) GTSRB (BWTP)   (d) GTSRB (GLBW)

Figure 7: The distance between potential trigger patterns and the original one used for training $w.r.t.$ the loss value on CIFAR-10 and GTSRB. The original trigger pattern is denoted by a 'red star'.

In particular, we design an adaptive optimization method to find promising synthesized patterns with the highest attack effectiveness and differences from the original trigger. Specifically, we repeat the trigger generation and adaptively adjust the $\mu$ based on the current trigger candidate untill we find the promising synthesized trigger in each inner maximization. The main pipeline of our method is shown in Figure 6. Please find more optimization details and its pseudocodes in Appendix E.

## 5 EXPERIMENTS

### 5.1 MAIN SETTINGS

**Baseline Selection.** In this paper, we compare our generalization-limited backdoor watermark (GLBW) with vanilla backdoor watermark (dubbed 'Vanilla') (Lin et al., 2021) and backdoor watermark with trigger penalty (BWTP) with standardized evaluation process.

**Settings for Model Watermarks.** Following settings in (Lin et al., 2021), we adopt the $3 \times 3$ white square located in the lower-right bottom as the original trigger on both CIFAR-10 and GTSRB. Specifically, we simply set $\lambda_1 = \lambda_2 = 1$ for BWTP and $\lambda_3 = \lambda_4 = 1$ for our GLBW. We use the $\ell_1$-norm as the distance measurement $d$ and exploit a modified version of the Sigmoid function as the probability activation function $p(\cdot)$ used in BWTP. Please refer to Appendix F for more details.

**Settings for the Generation of Potential Triggers.** In Section 3.2, we adopt neural cleanse (Wang et al., 2019) to generate potential triggers. In our GLBW, we also exploit a similar approach during the generation of synthesized triggers. To evaluate whether our method can truly decrease trigger generalization, we hereby adopt two other trigger synthesis methods, including TABOR (Guo et al., 2020) and pixel backdoor (Tao et al., 2022) to generate potential triggers for all watermarked DNNs. We generate 1,000 trigger candidates (with random initializations) in all cases. We only select those whose loss values are less than 1.5 times of that of the original trigger (as shown in the red line in Figure 7) as our potential trigger patterns. Please refer to Appendix F for more details.

**Evaluation Metrics.** We adopt benign accuracy (BA) and watermark success rate (WSR) to evaluate the effectiveness of model watermarks. To evaluate their trigger generalization, we exploit chamfer distance (dubbed 'Chamfer') and the percentage of effective potential triggers with low generalization (dubbed 'PLG'). The effective triggers whose IOU values are greater than the threshold (0.3 for CIFAR-10 and 0.18 for GTSRB) are regarded as with low generalization. In general, the smaller the chamfer distance and the larger the PLG, the lower the trigger generalization.

### 5.2 MAIN RESULTS

As shown in Table 5, both our BWTP and GLBW can obtain a sufficiently high WSR ($> 85\%$) on both datasets. In particular, our GLBW can significantly reduce trigger generalization in all cases (as shown in both Table 5 and Figure 7). For example, the chamfer distance (measured via neural cleanse and TABOR) of GLBW is more than 170 times smaller than that of the vanilla backdoor

Table 6: The average rank (based on the IOU) of SRV methods that is evaluated with our generalization-limited backdoor watermark on CIFAR-10 and GTSRB datasets.

| Dataset↓, SRV→ | BP | GBP | GCAM | GGCAM | OCC | FA | LIME |
|---|---|---|---|---|---|---|---|
| CIFAR-10 | 4.50 | 4.50 | 7.00 | 5.00 | 2.00 | 3.00 | 1.00 |
| GTSRB | 4.00 | 4.00 | 6.00 | 6.00 | 2.00 | 3.00 | 1.00 |

Table 7: Effects of the trigger size.

| Size | BA (%) | WSR (%) | Chamfer | PLG (%) |
|---|---|---|---|---|
| $3 \times 3$ | 78.33 | 93.68 | 23.04 | 100 |
| $4 \times 4$ | 83.20 | 91.58 | 67.74 | 100 |
| $5 \times 5$ | 87.81 | 91.79 | 90.44 | 100 |

Table 8: Effects of the target label.

| Label | BA (%) | WSR (%) | Chamfer | PLG (%) |
|---|---|---|---|---|
| 0 | 78.33 | 93.68 | 23.04 | 100 |
| 1 | 84.27 | 83.82 | 25.31 | 100 |
| 2 | 77.08 | 93.23 | 25.09 | 100 |

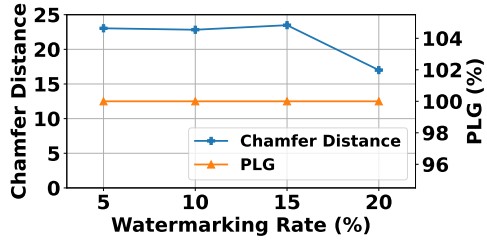

Figure 8: Effects of the watermarking rate.

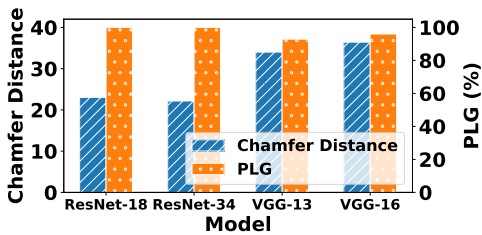

Figure 9: Effects of the Model Structure.

watermark on CIFAR-10. However, the BWTP may even increase the generalization on CIFAR-10, although it can reach remarkable performance on the GTSRB dataset. As we will show in Appendix G.3, it is mostly because its synthesized triggers are 'weak' with either a large loss value or are similar to the original trigger. These results demonstrate the effectiveness of our GLBW method.

We notice that both BWTP and GLBW may decrease the benign accuracy to some extent, compared to the vanilla backdoor watermark. However, this mild potential limitation will not hinder the usefulness (*i.e.*, more faithful evaluation) of our methods, since the watermarked DNNs are only used for evaluating SRV methods instead of for deployment (more details are in Appendix J). The average rank of SRV methods via our GLBW is shown in Table 6. As shown in this table, the results are consistent across datasets and therefore are more faithful and reliable. Please find the IOU of SRV methods via three discussed model watermarks under different settings in Appendix G.

### 5.3 THE EFFECTS OF KEY HYPER-PARAMETERS

We hereby verify whether our GLBW is still effective under different settings on CIFAR-10. The chamfer distance and PLG are measured via neural cleanse. More results are in Appendix H

**Effects of the Watermarking Rate $\gamma$.** As shown in Figure 8, the watermarking rate has minor effects on trigger generalization. In particular, our GLBW can reach high effectiveness (as shown in our appendix) and low generalization under different $\gamma$. It verifies the our effectiveness again.

**Effects of the Trigger Size.** As shown in Table 7, increasing the trigger size has minor effects to watermark effectiveness, although there are some fluctuations for BA and WSR. We notice that the chamfer distance raises with the increase in trigger size. However, it is due to the properties of this distance metric and does not mean an increase in trigger generalization. The PLG remains 100%.

**Effects of the Target Label $y_t$.** As shown in Table 8, our GLBW has promising performance under different $y_t$. The trigger generalization remains low with a small chamfer distance and 100% PLG.

**Effects of the Model Structure.** As shown in Figure 9, our GLBW reaches similar generalization ability, especially with similar model architectures.

## 6 CONCLUSION

In this paper, we revisited the evaluation methods of saliency-based representation visualization (SRV). We revealed the unreliable nature (due to the trigger generalization of backdoor watermarks) and implementation limitations of existing backdoor-based SRV evaluation methods. Based on these findings, we proposed a generalization-limited backdoor watermark, based on which we designed a more faithful XAI evaluation. We hope our work can provide a deeper understanding of XAI evaluation, to facilitate the design of more interpretable deep learning methods.

## ACKNOWLEDGMENTS

This work is supported in part by the National Natural Science Foundation of China under Grants (62302309, 62171248), Shenzhen Science and Technology Program under Grants (JCYJ20220818101014030, JCYJ20220818101012025), and the PCNL KEY Project (PCL2023AS6-1).

## ETHICS STATEMENT

This paper is the first attempt toward measuring and reducing trigger generalization of backdoor watermarks and its positive application in XAI evaluation. Accordingly, our paper has positive societal impacts in general. However, we notice that the adversaries may adopt our backdoor watermark with trigger penalty (BWTP) or generalization-limited backdoor watermark (GLBW) as backdoor attacks when they can control the training process of DNNs. However, these attacks are less harmful compared to existing backdoor attacks because they will significantly reduce the benign accuracy of attacked models and therefore can be noticed by victim model users. Besides, victim users can also exploit existing backdoor defenses, especially saliency-based methods, to detect or remove these backdoor attacks. Moreover, although we do not design a specified defense against BWTP or GLBW, people can still mitigate or even avoid the threats by only using trusted training resources.

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

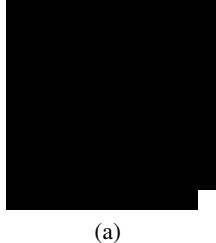 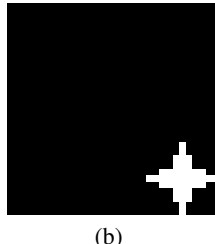

(a)                                    (b)

Figure 10: The watermark patterns used for standardized backdoor-based XAI evaluation.

APPENDIX

## A    SETTINGS FOR STANDARDIZE BACKDOOR-BASED XAI EVALUATION

**Datasets and Models.** We conduct experiments on two classical datasets, including CIFAR-10 (Krizhevsky, 2009) and GTSRB (Stallkamp et al., 2012), with ResNet-18 (He et al., 2016).

**Watermark Setup.** We adopt two watermark patterns: the square-like pattern (as shown in Figure 10(a)) and the compass-like pattern (as shown in Figure 10(b)). Based on these patterns, we generate four kinds of triggers: (a), (b), (c), and (d). Specifically, (a) and (c) are the triggers with the square-like pattern; (b) and (d) are triggers with the compass-like pattern; (a) and (b) are the triggers located in the lower-right corner; (c) and (d) are the triggers with random locations.

**Training Setup.** We adopt BadNets (Gu et al., 2019) to train our backdoored models based on the open-sourced Python toolbox—`BackdoorBox` (Li et al., 2023a). We train 8 backdoored models using four triggers (*i.e.*, (a), (b), (c), and (d)) on CIFAR-10 and GTSRB. We set the watermarking rate as 5% and the target label $y_t$ is set to 0 for all backdoor training. To train our backdoored models, the mini-batch is set to 128 for each iteration. We use the SGD optimizer by setting momentum=0.9 and weight_decay=5e-4. For CIFAR-10, the number of the training epochs is 200 and the learning rate is initialized as 0.1 and multiplied by 0.1 at the 150 and 180 epoch, respectively. For GTSRB, the number of the training epochs is 30 and the learning rate is initialized as 0.01 and multiplied by 0.1 at the 20th epoch. In particular, we add watermarks after performing the data augmentation.

**Computational Facilities.** We train each model with one NVIDIA RTX3090 GPU. For CIFAR-10, it costs about 40 minutes for training. For GTSRB, it costs about 15 minutes for training.

## B    SETTINGS FOR TRIGGER GENERALIZATION AND ITS INFLUENCES TO XAI EVALUATION

We select the four models watermarked with the triggers (a) and (c) on CIFAR-10 and GTSRB used in Section A for discussing trigger generalization. We generate 1,000 targeted universal adversarial perturbations (Wang et al., 2019) (with random initializations) regarding the target label. In particular, we used the neural cleanse (Wang et al., 2019) to generate all synthesized triggers, where the weight of the regularization term is set to $8 \times 10^{-5}$. The training dataset is used as the set of clean images for neural cleanse. We generate each synthesized trigger with 10 epochs.

## C    WHY DO WE (ONLY) NEED TO CONSIDER PATCH-BASED TRIGGERS?

People may worry about whether our method can truly lead more faithful evaluation since we only focus on patch-based attacks (*e.g.*, BadNets) and there are still many other backdoor attacks with different trigger patterns (Li et al., 2021b; Doan et al., 2021; Zeng et al., 2023), while people may also use complicated features in recognizing real images in practice.

Firstly, patch-based poisoned-label backdoor watermark is the most suitable and probably the only suitable method for backdoor-based SRV evaluation. Specifically, clean-label attacks where the adversaries only poison samples for the target class (Turner et al., 2019; Gao et al., 2023b;a; Zeng

Table 9: The effectiveness and generalization of backdoor watermarks on CIFAR-10 with different trigger types. The best results are marked in boldface.

| Type↓ | Method↓, Metric→ | BA | WSR | Chamfer | PLG |
|---|---|---|---|---|---|
| Blended | Vanilla | 97.55 | 91.79 | 61.59 | 94.60 |
| | GLBW | 90.10 | 88.03 | **9.81** | **100** |
| Additive | Vanilla | 97.60 | 95.01 | 66.47 | 95.00 |
| | GLBW | 94.93 | 91.92 | **26.42** | **100** |

et al., 2023) are not suitable since attacked DNNs may use both 'ground-truth' features and trigger features instead of just trigger features for predictions. Besides, the triggers of almost all existing non-patch-based attacks are with full image-size. We certainly don't want saliency ares are the whole image. Accordingly, these attacks are also not suitable for SRV evaluation.

Secondly, people categorize a given image usually based on its local regions instead of the whole image. For example, we only need to see the image areas of its 'head' to know it is a bird. These local regions are similar to the trigger patch used in our evaluation. Accordingly, results made by our method can be a good reference in real images for practice. This is probably the main reason why our baseline research (Lin et al., 2021) evaluated their method only with trigger patches.

Thirdly, it is impossible to faithfully evaluate the performance of SRV methods for complicated features (instead of simple trigger patches) since there are no ground-truth salience maps. Even a human expert cannot accurately mark the salience map for complicated features in most cases.

Fourthly, simple trigger patches are also features used by DNNs for their predictions. Accordingly, the evaluation of SRV methods on trigger features is the first and the most important step toward evaluating their general performance and is therefore of great significance.

In particular, one may argue that there are still two other patch-based triggers, including 'Blended' (*i.e.*, BadNets with trigger transparency) and patch-size additive trigger (dubbed 'Additive'), that we did not evaluated in our main experiments. Accordingly, we hereby conduct additional experiments on the CIFAR-10 dataset with them. As shown in Table 9, our GLBW is still highly effective.

## D   THE OPTIMIZATION PROCESS OF BWTP

Arguably, the most straightforward method is to synthesize the potential trigger pattern and penalize its effects based on its distance to the original one in each iteration during the training process. In general, if the potential trigger pattern is close to the ground-truth one, the predictions of watermarked DNNs to samples containing this pattern should be similar to the target label; Otherwise, their predictions should be similar to their ground-truth label. In this paper, we call this method as backdoor watermark with trigger penalty (BWTP), whose optimization process is as follows:

$$\min_{\boldsymbol{\theta}} \sum_{(\boldsymbol{x},y)\in\mathcal{D}} \underbrace{\mathcal{L}\left(f_{\boldsymbol{\theta}}\left(\boldsymbol{x}\right),y\right)}_{Benign\ Loss} + \lambda_1 \cdot \underbrace{\mathcal{L}\left(f_{\boldsymbol{\theta}}\left(\left(\boldsymbol{1}-\boldsymbol{m}\right)\otimes\boldsymbol{x}+\boldsymbol{m}\otimes\boldsymbol{t}\right),y_t\right)}_{Backdoor\ Loss} +$$
$$\lambda_2 \cdot \underbrace{\max_{\boldsymbol{m}',\boldsymbol{t}',|\boldsymbol{m}'|\leq T} \mathcal{L}'\left(f_{\boldsymbol{\theta}}\left(\left(\boldsymbol{1}-\boldsymbol{m}'\right)\otimes\boldsymbol{x}+\boldsymbol{m}'\otimes\boldsymbol{t}'\right),\hat{\boldsymbol{y}}_t+\hat{\boldsymbol{y}}\right)}_{Penalty\ Loss},$$
(1)

where $\mathcal{D}$ is the benign dataset, $\mathcal{L}$ and $\mathcal{L}'$ are the loss functions (*e.g.*, cross-entropy and mutual information), $\boldsymbol{t}$ is the original trigger pattern with mask $\boldsymbol{m}$, and $T$ is the threshold of the size of potential trigger mask $\boldsymbol{m}'$. $\hat{\boldsymbol{y}}_t = (1 - p(d(\boldsymbol{m},\boldsymbol{m}'))) \cdot \boldsymbol{y}_t$ and $\hat{\boldsymbol{y}} = p(d(\boldsymbol{m},\boldsymbol{m}')) \cdot \boldsymbol{y}$, where $\boldsymbol{y}$ is the one-hot representation of $y$, $d(\cdot,\cdot)$ is a distance metric (*e.g.*, chamfer distance), $p(\cdot)$ is a probability activation function to map the distance to a probability (the larger the distance, the higher the probability). $\lambda_1$ and $\lambda_2$ are two trade-off hyper-parameters for backdoor loss and penalty loss.

Similar to adversarial training (Madry et al., 2018), we alternately optimize the inner maximization and the outer minimization in each iteration. For the inner maximization, we solve the following formula to get a penalized trigger (with mask $\boldsymbol{m}'$ and pattern $\boldsymbol{t}'$):

$$\max_{\boldsymbol{m}',\boldsymbol{t}',|\boldsymbol{m}'|\leq T} \mathcal{L}'\left(f_{\boldsymbol{\theta}}\left((\boldsymbol{1}-\boldsymbol{m}')\otimes\boldsymbol{x}+\boldsymbol{m}'\otimes\boldsymbol{t}'\right),\hat{\boldsymbol{y}}_t+\hat{\boldsymbol{y}}\right). \tag{2}$$

For convenience, this formula can be written in an equivalent form:

$$\min_{\boldsymbol{m}',\boldsymbol{t}',|\boldsymbol{m}'|\leq T} -\mathcal{L}'\left(f_{\boldsymbol{\theta}}\left((\boldsymbol{1}-\boldsymbol{m}')\otimes\boldsymbol{x}+\boldsymbol{m}'\otimes\boldsymbol{t}'\right),\hat{\boldsymbol{y}}_t+\hat{\boldsymbol{y}}\right). \tag{3}$$

After that, this conditional minimization problem can be approximated as a non-conditional surrogate problem, as follow:

$$\min_{\boldsymbol{m}',\boldsymbol{t}'}\left[-\mathcal{L}'\left(f_{\boldsymbol{\theta}}\left((\boldsymbol{1}-\boldsymbol{m}')\otimes\boldsymbol{x}+\boldsymbol{m}'\otimes\boldsymbol{t}'\right),\hat{\boldsymbol{y}}_t+\hat{\boldsymbol{y}}\right)+\gamma\cdot\max(0,|\boldsymbol{m}'|-T)\right], \tag{4}$$

where $\gamma$ is the weight of the regularization term used for limiting the size of $\boldsymbol{m}'$.

We proposed an algorithm (as shown in Algorithm 1) to solve the aforementioned problem (4): Firstly, we randomly initialize $\boldsymbol{m}'_o$ and $\boldsymbol{t}'_o$ by sampling from the standard normal distribution. Secondly, we iteratively optimize the problem (4) on a benign dataset $\mathcal{D}$. In particular, we map the values of $\boldsymbol{m}'_o$ and $\boldsymbol{t}'_o$ to $[0,1]$ by a modification of tanh function in each iteration. After optimizing for several epochs, the final penalized trigger mask $\boldsymbol{m}'$ and pattern $\boldsymbol{t}'$ are obtained.

For the outer minimization, we adopt the penalized trigger mask and pattern generated from Algorithm 1 as $\boldsymbol{m}'$ and $\boldsymbol{t}'$. After that, we solve the following trying to reduce the trigger generalization of the watermarked DNN during its training process:

$$\min_{\boldsymbol{\theta}}\sum_{(\boldsymbol{x},y)\in\mathcal{D}} \underbrace{\mathcal{L}\left(f_{\boldsymbol{\theta}}\left(\boldsymbol{x}\right),y\right)}_{Benign\ Loss} + \lambda_1\cdot\underbrace{\mathcal{L}\left(f_{\boldsymbol{\theta}}\left((\boldsymbol{1}-\boldsymbol{m})\otimes\boldsymbol{x}+\boldsymbol{m}\otimes\boldsymbol{t}\right),y_t\right)}_{Backdoor\ Loss} +$$
$$\lambda_2\cdot\underbrace{\mathcal{L}'\left(f_{\boldsymbol{\theta}}\left((\boldsymbol{1}-\boldsymbol{m}')\otimes\boldsymbol{x}+\boldsymbol{m}'\otimes\boldsymbol{t}'\right),\hat{\boldsymbol{y}}_t+\hat{\boldsymbol{y}}\right)}_{Penalty\ Loss}. \tag{5}$$

We solve the aforementioned problem (5) with the standard SGD (Bottou, 2010), as shown in Algorithm 2. After several epochs, the final watermarked model $f_{\boldsymbol{\theta}}$ with BWTP is obtained.

## E  THE OPTIMIZATION PROCESS OF GLBW

As we show in the experiments, the previous BWTP suffers from relatively poor performance whose watermarked DNNs still have high trigger generalization in some cases. According to the intermediate results, this failure is mostly because the synthesized triggers are 'weak' with either a large loss value or are similar to the original trigger. Besides, the training of watermarked DNNs is unstable even when the synthesized triggers are promising. We argue that these limitations are due to the complexity of its penalty loss. As such, we propose to directly target the generation of synthesized triggers to make this process more controllable, inspired by neural cleanse which is a backdoor defense based on trigger synthesis (Wang et al., 2019). Its optimization is as follows:

$$\min_{\boldsymbol{\theta}}\sum_{(\boldsymbol{x},y)\in\mathcal{D}} \underbrace{\mathcal{L}\left(f_{\boldsymbol{\theta}}\left(\boldsymbol{x}\right),y\right)}_{Benign\ Loss} + \lambda_3\cdot\underbrace{\mathcal{L}\left(f_{\boldsymbol{\theta}}\left((\boldsymbol{1}-\boldsymbol{m})\otimes\boldsymbol{x}+\boldsymbol{m}\otimes\boldsymbol{t}\right),y_t\right)}_{Backdoor\ Loss} +$$
$$\lambda_4\cdot\underbrace{\max_{\boldsymbol{m}',\boldsymbol{t}',|\boldsymbol{m}'\cap\boldsymbol{m}|\leq\tau\cdot|\boldsymbol{m}|} -\left\{\mathcal{L}\left(f_{\boldsymbol{\theta}}\left((\boldsymbol{1}-\boldsymbol{m}')\otimes\boldsymbol{x}+\boldsymbol{m}'\otimes\boldsymbol{t}'\right),y_t\right)+\mu\cdot|\boldsymbol{m}'|\right\}}_{Generalization\ Loss} \tag{6}$$

where $\mu$ is a hyper-parameter trading off its size and effectiveness for synthesized trigger, $\lambda_3$ & $\lambda_4$ are two hyper-parameters for losses, $\tau$ is the overlap threshold between synthesized and original

---

**Algorithm 1** Generating the penalized trigger.

---

**Input:** The benign dataset $\mathcal{D}$; The current model $f_{\boldsymbol{\theta}}$; Target label $y_t$; The mask $\boldsymbol{m}$ and pattern $\boldsymbol{t}$ of the original trigger; Hyper-parameters $T$ and $\gamma$; The number of epochs for generating the penalized trigger $N_{gen}$;
**Output:** The mask $\boldsymbol{m}'$ and pattern $\boldsymbol{t}'$ of the penalized trigger;

1: **function** GET_PENALIZED_TRIGGER($\mathcal{D}, f_{\boldsymbol{\theta}}, y_t, \boldsymbol{m}, \boldsymbol{t}, T, \gamma, N_{gen}$)
2: $\quad \boldsymbol{m}'_o \sim \mathcal{N}(0, 1)$
3: $\quad \boldsymbol{t}'_o \sim \mathcal{N}(0, 1)$
4: $\quad$ **for** $n = 1$ to $N_{gen}$ **do**
5: $\quad\quad$ **for** each mini-batch $(\boldsymbol{x}, y) \in \mathcal{D}$ **do**
6: $\quad\quad\quad \boldsymbol{m}' \leftarrow \frac{\tanh(\boldsymbol{m}'_o)+1}{2}$ $\quad\quad \triangleright$ Map the values of $\boldsymbol{m}'_o$ to $[0,1]$
7: $\quad\quad\quad \boldsymbol{t}' \leftarrow \frac{\tanh(\boldsymbol{t}'_o)+1}{2}$ $\quad\quad \triangleright$ Map the values of $\boldsymbol{t}'_o$ to $[0,1]$
8: $\quad\quad\quad \boldsymbol{g}_{\boldsymbol{m}'_o} \leftarrow \nabla_{\boldsymbol{m}'_o} \left[ -\mathcal{L}'(f_{\boldsymbol{\theta}}((\mathbf{1} - \boldsymbol{m}') \otimes \boldsymbol{x} + \boldsymbol{m}' \otimes \boldsymbol{t}'), \hat{\boldsymbol{y}}_t + \hat{\boldsymbol{y}}) + \gamma \cdot \max(0, |\boldsymbol{m}'| - T) \right]$
9: $\quad\quad\quad \boldsymbol{g}_{\boldsymbol{t}'_o} \leftarrow \nabla_{\boldsymbol{t}'_o} \left[ -\mathcal{L}'(f_{\boldsymbol{\theta}}((\mathbf{1} - \boldsymbol{m}') \otimes \boldsymbol{x} + \boldsymbol{m}' \otimes \boldsymbol{t}'), \hat{\boldsymbol{y}}_t + \hat{\boldsymbol{y}}) + \gamma \cdot \max(0, |\boldsymbol{m}'| - T) \right]$
10: $\quad\quad\quad$ Use Adam optimizer to update $\boldsymbol{m}'_o$ and $\boldsymbol{t}'_o$ with gradients $\boldsymbol{g}_{\boldsymbol{m}'_o}$ and $\boldsymbol{g}_{\boldsymbol{t}'_o}$
11: $\quad\quad$ **end for**
12: $\quad$ **end for**
13: $\quad$ **return** $\boldsymbol{m}', \boldsymbol{t}'$
14: **end function**

---

**Algorithm 2** Watermarking DNNs with BWTP.

---

**Input:** The benign dataset $\mathcal{D}$; Model $f_{\boldsymbol{\theta}}$; Target label $y_t$; The mask $\boldsymbol{m}$ and pattern $\boldsymbol{t}$ of the original trigger; Hyper-parameters $\lambda_1, \lambda_2, T, \gamma$; The number of the training epochs $N_{train}$ and the number of epochs for generating the penalized trigger $N_{gen}$;
**Output:** BWTP-watermarked DNN $f_{\boldsymbol{\theta}}$;

1: **function** BWTP($\mathcal{D}, f_{\boldsymbol{\theta}}, y_t, \boldsymbol{m}, \boldsymbol{t}, \lambda_1, \lambda_2, T, \gamma, N_{train}, N_{gen}$)
2: $\quad$ **for** $n \leftarrow 1$ **to** $N_{train}$ **do**
3: $\quad\quad \boldsymbol{m}', \boldsymbol{t}' \leftarrow$ GET_PENALTY_TRIGGER($\mathcal{D}, f_{\boldsymbol{\theta}}, y_t, \boldsymbol{m}, \boldsymbol{t}, T, \gamma, N_{gen}$)
4: $\quad\quad$ **for** each mini-batch $(\boldsymbol{x}, y) \in \mathcal{D}$ **do**
5: $\quad\quad\quad \mathcal{L}_{benign} \leftarrow \mathcal{L}(f_{\boldsymbol{\theta}}(\boldsymbol{x}), y)$
6: $\quad\quad\quad \mathcal{L}_{backdoor} \leftarrow \mathcal{L}(f_{\boldsymbol{\theta}}((\mathbf{1} - \boldsymbol{m}) \otimes \boldsymbol{x} + \boldsymbol{m} \otimes \boldsymbol{t}), y_t)$
7: $\quad\quad\quad \mathcal{L}_{penalty} \leftarrow \mathcal{L}'(f_{\boldsymbol{\theta}}((\mathbf{1} - \boldsymbol{m}') \otimes \boldsymbol{x} + \boldsymbol{m}' \otimes \boldsymbol{t}'), \hat{\boldsymbol{y}}_t + \hat{\boldsymbol{y}})$
8: $\quad\quad\quad \boldsymbol{g}_{\boldsymbol{\theta}} \leftarrow \nabla_{\boldsymbol{\theta}} \left( \mathcal{L}_{benign} + \lambda_1 \cdot \mathcal{L}_{backdoor} + \lambda_2 \cdot \mathcal{L}_{penalty} \right)$
9: $\quad\quad\quad$ Use SGD optimizer to update $\boldsymbol{\theta}$ with gradient $\boldsymbol{g}_{\boldsymbol{\theta}}$
10: $\quad\quad$ **end for**
11: $\quad$ **end for**
12: $\quad$ **return** $f_{\boldsymbol{\theta}}$
13: **end function**

---

triggers. Notice that the $y$ and $y_t$ contained in GLBW are different from $\hat{\boldsymbol{y}}$ and $\hat{\boldsymbol{y}}_t$ used in the previous BWTP since these two methods are fundamentally different.

To solve the aforementioned problem (6), we alternately optimize the inner maximization and the outer minimization in each iteration. For the inner maximization, we solve the following maximization to get a promising synthesized trigger:

$$\max_{\boldsymbol{m}', \boldsymbol{t}', |\boldsymbol{m}' \cap \boldsymbol{m}| \leq \tau \cdot |\boldsymbol{m}|} - \left[ \mathcal{L}\left( f_{\boldsymbol{\theta}}\left( (\mathbf{1} - \boldsymbol{m}') \otimes \boldsymbol{x} + \boldsymbol{m}' \otimes \boldsymbol{t}' \right), y_t \right) + \mu \cdot |\boldsymbol{m}'| \right]. \quad (7)$$

For convenience, this problem can be written in an equivalent form:

$$\min_{\boldsymbol{m}', \boldsymbol{t}', |\boldsymbol{m}' \cap \boldsymbol{m}| \leq \tau \cdot |\boldsymbol{m}|} \left[ \mathcal{L}\left( f_{\boldsymbol{\theta}}\left( (\mathbf{1} - \boldsymbol{m}') \otimes \boldsymbol{x} + \boldsymbol{m}' \otimes \boldsymbol{t}' \right), y_t \right) + \mu \cdot |\boldsymbol{m}'| \right]. \quad (8)$$

In particular, we design an adaptive optimization algorithm (as shown in Algorithm 3) to solve the aforementioned problem (8): At the beginning, we initialize $init\_flag$ to $False$ and $\mu$ to $\mu_0$. $init\_flag$ is flag used to identify whether to force $\boldsymbol{m}'$ to be different from $\boldsymbol{m}$ by the special initialization of $\boldsymbol{m}'_o$. $\mu$ is an adaptive parameter to limit the size of the synthesized trigger mask $\boldsymbol{m}'$. After

---

**Algorithm 3** Generating the desired synthesized trigger.

---

**Input:** The benign dataset $\mathcal{D}$; The current model $f_{\boldsymbol{\theta}}$; Target label $y_t$; The mask $\boldsymbol{m}$ and trigger pattern $\boldsymbol{t}$ of the original trigger; Hyper-parameters $\mu_0$ and $\tau$; The number of epochs for generating the potential trigger $N_{gen}$;
**Output:** The mask $\boldsymbol{m}'$ and its pattern $\boldsymbol{t}'$ of the promising synthesized trigger;

---

1: **function** GET_PROMISING_TRIGGER($\mathcal{D}, f_{\boldsymbol{\theta}}, y_t, \boldsymbol{m}, \boldsymbol{t}, \mu_0, \tau, N_{gen}$)
2:     $init\_flag \leftarrow False$
3:     $\mu \leftarrow \mu_0$
4:     **loop**
5:         $\boldsymbol{m}'_o \sim \mathcal{N}(0, 1)$
6:         $\boldsymbol{t}'_o \sim \mathcal{N}(0, 1)$
7:         **if** $init\_flag = True$ **then**
8:             $\boldsymbol{m}'_{o_{i,j}} \leftarrow -\infty$ where $\boldsymbol{m}_{i,j} = 1$
9:         **end if**
10:        **for** $n = 1$ to $N_{gen}$ **do**
11:           **for** each mini-batch $\boldsymbol{x} \in \mathcal{D}$ **do**
12:               $\boldsymbol{m}' \leftarrow \frac{\tanh(\boldsymbol{m}'_o)+1}{2}$       ▷ Map the values of $\boldsymbol{m}'_o$ to $[0, 1]$
13:               $\boldsymbol{t}' \leftarrow \frac{\tanh(\boldsymbol{t}'_o)+1}{2}$        ▷ Map the values of $\boldsymbol{t}'_o$ to $[0, 1]$
14:               $\boldsymbol{g}_{\boldsymbol{m}'_o} \leftarrow \nabla_{\boldsymbol{m}'_o}[\mathcal{L}(f_{\boldsymbol{\theta}}((\boldsymbol{1} - \boldsymbol{m}') \otimes \boldsymbol{x} + \boldsymbol{m}' \otimes \boldsymbol{t}'), y_t) + \mu \cdot |\boldsymbol{m}'|]$
15:               $\boldsymbol{g}_{\boldsymbol{t}'_o} \leftarrow \nabla_{\boldsymbol{t}'_o}[\mathcal{L}(f_{\boldsymbol{\theta}}((\boldsymbol{1} - \boldsymbol{m}') \otimes \boldsymbol{x} + \boldsymbol{m}' \otimes \boldsymbol{t}'), y_t) + \mu \cdot |\boldsymbol{m}'|]$
16:               Use Adam optimizer to update $\boldsymbol{m}'_o$ and $\boldsymbol{t}'_o$ with gradients $\boldsymbol{g}_{\boldsymbol{m}'_o}$ and $\boldsymbol{g}_{\boldsymbol{t}'_o}$
17:           **end for**
18:        **end for**
19:        $loss_{\boldsymbol{m}} \leftarrow \frac{1}{|\mathcal{D}|} \sum_{\boldsymbol{x} \in \mathcal{D}} \mathcal{L}(f_{\boldsymbol{\theta}}((\boldsymbol{1} - \boldsymbol{m}) \otimes \boldsymbol{x} + \boldsymbol{m} \otimes \boldsymbol{t}), y_t)$
20:        $loss_{\boldsymbol{m}'} \leftarrow \frac{1}{|\mathcal{D}|} \sum_{\boldsymbol{x} \in \mathcal{D}} \mathcal{L}(f_{\boldsymbol{\theta}}((\boldsymbol{1} - \boldsymbol{m}') \otimes \boldsymbol{x} + \boldsymbol{m}' \otimes \boldsymbol{t}'), y_t)$
21:        **if** $|\boldsymbol{m}'| > |\boldsymbol{m}| \cdot 2$ **then**
22:           $\mu \leftarrow \mu \cdot \frac{|\boldsymbol{m}'|}{|\boldsymbol{m}|}$    ▷ Adjust $\mu$ adaptively
23:           **continue**
24:        **else if** $|\boldsymbol{m}'| < |\boldsymbol{m}| \cdot 0.6$ **and** $loss_{\boldsymbol{m}'} > loss_{\boldsymbol{m}}$ **then**
25:           $\mu \leftarrow \mu \cdot 0.618$   ▷ Adjust $\mu$ adaptively
26:           **continue**
27:        **else if** $loss_{\boldsymbol{m}'} > 1.8 \cdot loss_{\boldsymbol{m}}$ **or** $loss_{\boldsymbol{m}'} + \mu \cdot |\boldsymbol{m}'| > 1.5 \cdot (loss_{\boldsymbol{m}} + \mu \cdot |\boldsymbol{m}|)$ **then**
28:           **continue**      ▷ We get a bad trigger candidate and run the algorithm again to find a better one.
29:        **else if** $|\boldsymbol{m}' \cap \boldsymbol{m}| > \tau \cdot |\boldsymbol{m}|$ **then**
30:           $init\_flag \leftarrow True$
31:           **continue**   ▷ We get a trigger candidate that is too similar to the original trigger $\boldsymbol{m}$ and run the algorithm again to find a different one.
32:        **end if**
33:        **return** $\boldsymbol{m}', \boldsymbol{t}'$
34:     **end loop**
35: **end function**

---

initializing $init\_flag$ and $\mu_0$, the program enters a loop to generate potential synthesized triggers adaptively. In the loop, the first step is randomly initializing $\boldsymbol{m}'_o$ and $\boldsymbol{t}'_o$ by sampling from the standard normal distribution. If $init\_flag$ is $True$, we set $\boldsymbol{m}'_{o_{i,j}}$ to $-\infty$ where $\boldsymbol{m}_{i,j} = 1$. This is to force $\boldsymbol{m}'$ to be different from $\boldsymbol{m}$ (that is $\boldsymbol{m}' \cap \boldsymbol{m} = \boldsymbol{0}$). After that, we iteratively optimize the the problem (8) on the benign dataset $\mathcal{D}$. After optimizing for several epochs, we get a trigger candidate mask $\boldsymbol{m}'$ and its corresponding pattern $\boldsymbol{t}'$. Then, we calculate the loss of the original trigger mask $\boldsymbol{m}$ with target label $y_t$ (dubbed as $loss_{\boldsymbol{m}}$) and the loss of the candidate trigger mask $\boldsymbol{m}'$ with target label $y_t$ (dubbed as $loss_{\boldsymbol{m}'}$). After that, we will check the quality of the trigger candidate. The first check is whether the size of candidate trigger mask $\boldsymbol{m}'$ is too big. If $|\boldsymbol{m}'| > 2 \cdot |\boldsymbol{m}|$, we increase $\mu$ by multiplying $\frac{|\boldsymbol{m}'|}{|\boldsymbol{m}|}$ to limit the size of $\boldsymbol{m}'$ and then continue the loop to generate another trigger candidate; The second check is whether the size of candidate trigger mask $\boldsymbol{m}'$ is too

---

**Algorithm 4** Watermarking DNNs with GLBW.

---

**Input:** The benign dataset $\mathcal{D}$; The model $f_{\boldsymbol{\theta}}$; Target label $y_t$; The mask $\boldsymbol{m}$ and pattern $\boldsymbol{t}$ of the original trigger; Hyper-parameters $\lambda_3, \lambda_4, \mu_0, \tau$; The number of the training epochs $N_{train}$ and the number of epochs for generating the penalized trigger $N_{gen}$;
**Output:** A model with generalization-limited backdoor watermark $f_{\boldsymbol{\theta}}$;

1: **function** GLBW($\mathcal{D}, f_{\boldsymbol{\theta}}, y_t, \boldsymbol{m}, \boldsymbol{t}, \lambda_3, \lambda_4, \mu_0, \tau, N_{train}, N_{gen}$)
2:     **for** $n \leftarrow 1$ **to** $N_{train}$ **do**
3:         $\boldsymbol{m}', \boldsymbol{t}' \leftarrow$ GET_PROMISING_TRIGGER($\mathcal{D}, f_{\boldsymbol{\theta}}, y_t, \boldsymbol{m}, \boldsymbol{t}, \mu_0, \tau, N_{gen}$)
4:         **for** each mini-batch $(\boldsymbol{x}, y) \in \mathcal{D}$ **do**
5:             $\mathcal{L}_{benign} \leftarrow \mathcal{L}(f_{\boldsymbol{\theta}}(\boldsymbol{x}), y)$
6:             $\mathcal{L}_{backdoor} \leftarrow \mathcal{L}(f_{\boldsymbol{\theta}}((\boldsymbol{1} - \boldsymbol{m}) \otimes \boldsymbol{x} + \boldsymbol{m} \otimes \boldsymbol{t}), y_t)$
7:             $\mathcal{L}_{generalization} \leftarrow \mathcal{L}(f_{\boldsymbol{\theta}}((\boldsymbol{1} - \boldsymbol{m}') \otimes \boldsymbol{x} + \boldsymbol{m}' \otimes \boldsymbol{t}'), y)$
8:             $\boldsymbol{g_\theta} \leftarrow \nabla_{\boldsymbol{\theta}} \left( \mathcal{L}_{benign} + \lambda_3 \cdot \mathcal{L}_{backdoor} + \lambda_4 \cdot \mathcal{L}_{generalization} \right)$
9:             Use SGD optimizer to update $\boldsymbol{\theta}$ with gradient $\boldsymbol{g_\theta}$
10:        **end for**
11:     **end for**
12:     **return** $f_{\boldsymbol{\theta}}$
13: **end function**

---

small. If $|\boldsymbol{m}'| < 0.6 \cdot |\boldsymbol{m}|$ and $loss_{\boldsymbol{m}'} > loss_{\boldsymbol{m}}$, that is the trigger mask is too small and with a large loss, we decrease $\mu$ by multiplying 0.618 to relax the size of $\boldsymbol{m}'$ and then continue the loop to generate another trigger candidate; The third check is whether the candidate trigger is effective. If $loss_{\boldsymbol{m}'} > 1.8 \cdot loss_{\boldsymbol{m}}$ or $loss_{\boldsymbol{m}'} + \mu \cdot |\boldsymbol{m}'| > 1.5 \cdot (loss_{\boldsymbol{m}} + \mu \cdot |\boldsymbol{m}|)$, we think the trigger candidate is too bad and cannot effectively activate the backdoor, then continue the loop to generate another better trigger candidate; The fourth check is whether the candidate trigger is similar to the original trigger. If $|\boldsymbol{m}' \cap \boldsymbol{m}| > \tau \cdot |\boldsymbol{m}|$, we get a trigger candidate that is too similar to the original trigger $\boldsymbol{m}$, we set $init\_flag$ to $True$ and run the algorithm again to find a different one. If the trigger candidate passes all four checks, it is a promising synthesized trigger and we mark its mask $\boldsymbol{m}'$ and pattern $\boldsymbol{t}'$.

For the outer minimization, we use the promising synthesized trigger mask and pattern generated from Algorithm 3 as $\boldsymbol{m}'$ and $\boldsymbol{t}'$. After that, we solve the following problem to train a generalization-limited watermarked model:

$$\min_{\boldsymbol{\theta}} \sum_{(\boldsymbol{x},y) \in \mathcal{D}} \underbrace{\mathcal{L}\left(f_{\boldsymbol{\theta}}\left(\boldsymbol{x}\right), y\right)}_{Benign\ Loss} + \lambda_3 \cdot \underbrace{\mathcal{L}\left(f_{\boldsymbol{\theta}}\left((\boldsymbol{1} - \boldsymbol{m}) \otimes \boldsymbol{x} + \boldsymbol{m} \otimes \boldsymbol{t}\right), y_t\right)}_{Backdoor\ Loss} +$$
$$\lambda_4 \cdot \underbrace{\mathcal{L}\left(f_{\boldsymbol{\theta}}\left((\boldsymbol{1} - \boldsymbol{m}') \otimes \boldsymbol{x} + \boldsymbol{m}' \otimes \boldsymbol{t}'\right), y_t\right)}_{Generalization\ Loss}, \tag{9}$$

We solve the aforementioned problem (9) with the standard SGD (Bottou, 2010), as shown in Algorithm 4. After several epochs, the final GLBW-watermarked model $f_{\boldsymbol{\theta}}$ is obtained.

## F  DETAILED SETTINGS FOR MAIN EXPERIMENTS

**General Settings for Backdoor Watermarks.** We adopt the trigger (a) used in Section A as the original trigger pattern for all model watermarks on both CIFAR-10 and GTSRB. The watermarking rate and the target label are also the same as those used in Section A.

**Settings for Vanilla Backdoor Watermark.** The settings are the same as those used in Section A.

**Settings for BWTP.** We set the the threshold of the size of potential trigger mask $T$ as 18, the number of epochs for generating each potential trigger $N_{gen}$ as 10, $\lambda_1 = \lambda_2 = \gamma = 1$, and the distance measurement $d(\boldsymbol{m}, \boldsymbol{m}') = |\boldsymbol{m} - \boldsymbol{m}'|$. In particular, we use a modified version of the Sigmoid function as the probability activation function $p(\cdot)$ (as shown in Figure 11). When $d(\boldsymbol{m}, \boldsymbol{m}')$ is very large, the probability $p(d(\boldsymbol{m}, \boldsymbol{m}')) \approx 1$ and the penalty label $\hat{\boldsymbol{y}}_t + \hat{\boldsymbol{y}} \approx \boldsymbol{y}$; When $d(\boldsymbol{m}, \boldsymbol{m}')$ is very small, the probability activation $p(d(\boldsymbol{m}, \boldsymbol{m}')) \approx 0$ and the penalty label $\hat{\boldsymbol{y}}_t + \hat{\boldsymbol{y}} \approx \boldsymbol{y}_t$.

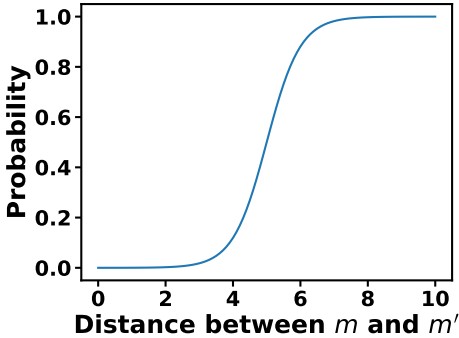

Figure 11: The graph of the probability activation function $p(\cdot)$.

**Settings for GLBW.** We set the hyper-parameters $\lambda_3 = \lambda_4 = 1, \mu_0 = 0.001, \tau = 0.05$ and the number of epochs for generating each potential trigger $N_{gen}$ as 15.

**General Settings for Trigger Synthesis.** We adopt three trigger synthesis methods, including neural cleanse (Wang et al., 2019), TABOR (Guo et al., 2020), and pixel backdoor (Tao et al., 2022) to generate potential triggers of watermarked DNNs. We generate 1,000 trigger candidates (with random initializations) in all cases. We only select those whose loss values are less than 1.5 times of that of the original trigger as our potential triggers.

**Evaluation Metrics.** Following the most classical settings, we adopt benign accuracy (BA) and watermark success rate (WSR) to evaluate the effectiveness of model watermarks. To evaluate their trigger generalization, we exploit chamfer distance (dubbed 'Chamfer') and the percentage of effective potential triggers with low generalization (dubbed 'PLG'). The effective triggers whose IOU values are greater than the threshold (0.3 for CIFAR-10 and 0.18 for GTSRB) are regarded as with low generalization. In general, the smaller the chamfer distance and the larger the PLG, the lower the position generalization of a watermark.

**The Visualization of Different Types of Trigger Candidates and The Definition of PLG.** We visualize different trigger candidates for better understanding (as shown in Figure 12). Triggers in the first line are effective candidates with a high IOU value (*i.e.*, effective triggers with low generalization). These triggers have high watermark success rate (WSR) and low backdoor loss that can activate the backdoor of watermarked DNNs successfully; Triggers in the second line are effective candidates with a low IOU value (*i.e.*, effective triggers with high generalization). These potential triggers have high WSR and low backdoor loss that can activate the backdoor of watermarked DNNs successfully, whereas they are different from the original trigger pattern; Triggers in the third line are ineffective trigger candidates with low WSR and high backdoor loss that can not activate model backdoors. Accordingly, they should not be regarded as potential triggers but targeted universal adversarial perturbations. We do not include them in the evaluation of trigger generalization. As such, the caculation of PLG can be written as follows:

$$PLG = \frac{\text{the number of the effective triggers with high IOU}}{\text{the number of all effective triggers}}.$$

**Settings for Neurl Cleanse.** We set the number of training epochs as 10 for the vanilla watermark and 3 for others. On the CIFAR-10 dataset, we set the weight of the regularization term $\lambda = 8 \times 10^{-5}$ for the vanilla watermark, $\lambda = 6 \times 10^{-2}$ for BWTP, and $\lambda = 9.315 \times 10^{-2}$ for GLBW. On the GTSRB dataset, we set $\lambda = 8 \times 10^{-5}$ for the vanilla watermark, $\lambda = 9 \times 10^{-2}$ for BWTP, and $\lambda = 9 \times 10^{-2}$ for GLBW. We adopt different settings to achieve the best synthesis performance.

**Settings for TABOR.** We set the number of training epochs as 2. On the CIFAR-10 dataset, we set the weights of the TABOR regularization terms $\lambda_1 = 8 \times 10^{-4}, \lambda_2 = 10^{-4}, \lambda = 10^{-5}$ for the vanilla watermark, $\lambda_1 = 5 \times 10^{-2}, \lambda_2 = 5 \times 10^{-3}, \lambda = 10^{-5}$ for BWTP, and $\lambda_1 = 7.5 \times 10^{-2}, \lambda_2 = 7.5 \times 10^{-3}, \lambda = 10^{-5}$ for GLBW. On the GTSRB dataset, we set $\lambda_1 = 8 \times 10^{-4}, \lambda_2 = 10^{-4}, \lambda = 10^{-5}$ for the vanilla watermark, $\lambda_1 = 8 \times 10^{-2}, \lambda_2 = 8 \times 10^{-3}, \lambda = 10^{-5}$ for BWTP, and $\lambda_1 = 7.2 \times 10^{-2}, \lambda_2 = 7.2 \times 10^{-3}, \lambda = 10^{-5}$ for GLBW. We adopt different settings in different cases to achieve the best synthesis performance.

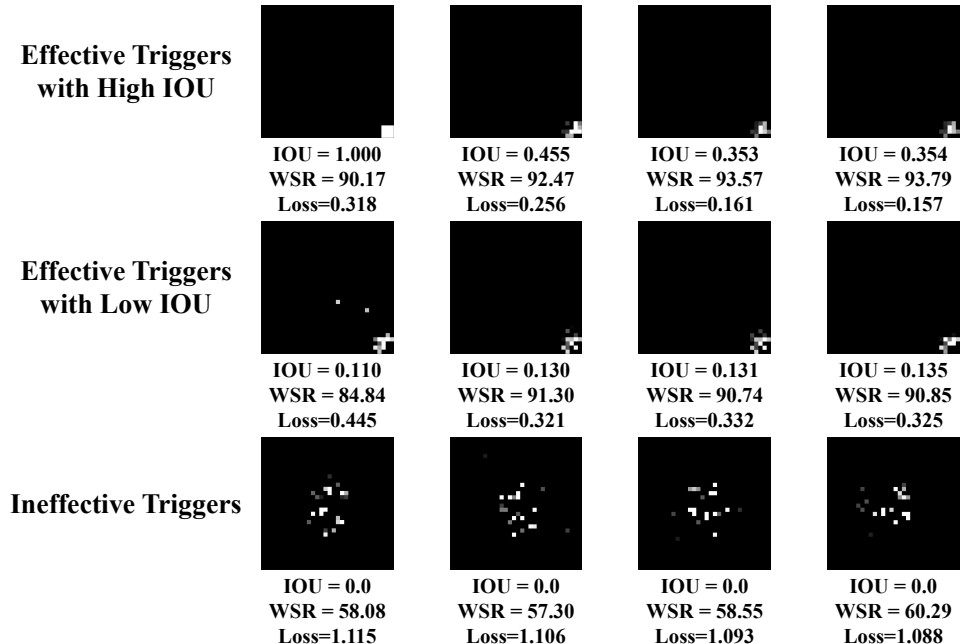

Figure 12: The example of synthesized trigger candidates generated by TABOR towards models watermarked by our GLBW on the GTSRB dataset.

Table 10: The effectiveness and generalization of model watermarks on CIFAR-10 and GTSRB datasets. '# All' indicates the number of all triggers while '# P' denotes that of potential triggers. '–' denotes that the trigger synthesis method can not find any potential trigger in this case.

| Dataset↓ | Trigger Synthesis Method→ | | | Neural Cleanse | | TABOR | | Pixel Backdoor | |
|---|---|---|---|---|---|---|---|---|---|
| | Metric→ Watermark↓ | BA | WSR | # All | # P | $\ell_1$-norm | # P | $\ell_1$-norm | # P | $\ell_1$-norm |
| CIFAR-10 | Vanilla | 91.73 | 97.39 | 1000 | 1000 | 20.61 | 1000 | 21.17 | 1000 | 14.912 |
| | BWTP | 85.51 | 89.16 | 1000 | 424 | 25.08 | 314 | 26.83 | 0 | - |
| | GLBW | 78.33 | 93.68 | 1000 | 63 | 6.20 | 12 | 6.71 | 6 | 6.434 |
| GTSRB | Vanilla | 97.57 | 94.52 | 1000 | 1000 | 9.48 | 638 | 9.80 | 1000 | 10.113 |
| | BWTP | 88.36 | 91.61 | 1000 | 38 | 5.79 | 6 | 5.47 | 11 | 6.673 |
| | GLBW | 93.89 | 90.17 | 1000 | 428 | 7.43 | 100 | 9.39 | 86 | 8.451 |

**Settings for Pixel Backdoor.** We set the number of training epochs as 10. On the CIFAR-10 dataset, we set the weight of the regularization term $\alpha = 5 \times 10^{-4}$ for the vanilla watermark, $\alpha = 0.5$ for BWTP, and $\alpha = 2$ for GLBW. On the GTSRB dataset, we set $\alpha = 5 \times 10^{-4}$ for the vanilla watermark, $\alpha = 0.7$ for BWTP, and $\alpha = 1$ for GLBW. We adopt different settings in different cases to achieve the best synthesis performance towards all watermarked models.

# G    ADDITIONAL RESULTS OF MAIN EXPERIMENTS

In this section, we provide additional results that are omitted in the main manuscript.

## G.1    THE GENERALIZATION OF MODEL WATERMARKS

In the main manuscript, we adopt chamfer distance and PLG to measure the trigger generalization. In this section, we provide the results under $\ell_1$-norm. Besides, we provide the number of all synthesized triggers (dubbed '# All') and potential triggers (dubbed '# P').

As shown in Table 10, our GLBW can significantly reduce the trigger generalization compared to the vanilla backdoor watermark in all cases whereas BWTP may even increase it in some cases.

Table 11: The evaluation (IOU) of SRV methods based on our generalization-limited backdoor watermark with the standardized process on the CIFAR-10 dataset and the GTSRB dataset. **(a)&(c)**: triggers with the square-like pattern; **(b)&(d)**: triggers with the compass-like pattern; **(a)&(b)**: triggers located in the lower-right corner; **(c)&(d)**: triggers with random locations.

| Dataset↓ | SRV→ Trigger↓ | BP | GBP | GCAM | GGCAM | OCC | FA | LIME |
|---|---|---|---|---|---|---|---|---|
| CIFAR-10 | (a) | 0.143 | 0.143 | 0.000 | 0.003 | 0.603 | 0.543 | 0.990 |
| | (b) | 0.148 | 0.148 | 0.045 | 0.215 | 0.621 | 0.583 | 0.997 |
| | (c) | 0.116 | 0.116 | 0.000 | 0.138 | 0.927 | 0.652 | 0.964 |
| | (d) | 0.260 | 0.260 | 0.000 | 0.134 | 0.679 | 0.589 | 0.994 |
| GTSRB | (a) | 0.169 | 0.169 | 0.009 | 0.009 | 0.938 | 0.901 | 0.998 |
| | (b) | 0.247 | 0.247 | 0.040 | 0.040 | 0.693 | 0.685 | 0.985 |
| | (c) | 0.082 | 0.082 | 0.009 | 0.009 | 0.964 | 0.495 | 0.985 |
| | (d) | 0.096 | 0.096 | 0.040 | 0.040 | 0.666 | 0.520 | 0.952 |

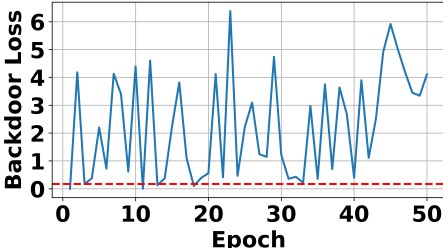

Figure 13: The backdoor loss of the synthesized trigger $w.r.t.$ training epoch of BWTP on CIFAR-10. The red dash line denotes the loss of the original trigger.

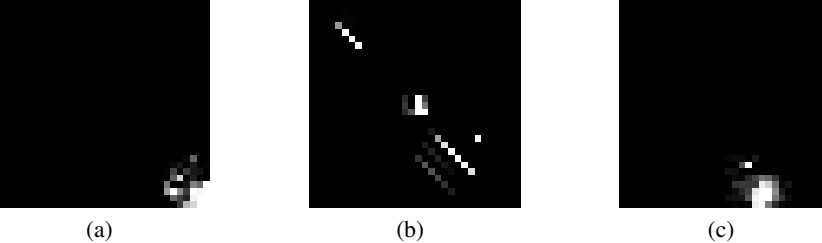

|  (a)  |  (b)  |  (c)  |

Figure 14: The effective synthesized triggers generated by BWTP on the CIFAR-10 dataset.

## G.2    THE IOU OF SRV METHODS WITH OUR GLBW

The IOU of SRV methods calculated based on our GLBW is shown in Table 11. The results show that our GLBW can lead to more consistent and stable evaluation and therefore is more faithful.

## G.3    A CLOSER LOOK TO THE FAILURE OF BWTP

In this section, we explore why our BWTP method may lead to poor performance on CIFAR-10 by visualizing the training process of BWTP. Specifically, we present the backdoor loss of each synthesized trigger with respect to the training epoch on the CIFAR-10 dataset.

As shown in Figure 13, the loss values of synthesized triggers are significantly larger than that of the original trigger. In other words, most of them are ineffective and cannot activate the backdoor in the watermarked model. Even among the few valid synthesized triggers, there is still a trigger pattern that is similar to the original trigger (as shown in Figure 14). Accordingly, penalizing these synthesized triggers has a minor benefit to reduce trigger generalization.

Table 12: Effects of the hyper-parameter $\lambda_3$.

| $\lambda_3$ | BA (%) | WSR (%) | Chamfer | PLG (%) |
|---|---|---|---|---|
| 0.6 | 78.35 | 93.15 | 23.13 | 100 |
| 0.8 | 81.49 | 92.45 | 17.38 | 100 |
| 1.0 | 78.33 | 93.68 | 23.04 | 100 |
| 1.2 | 82.63 | 93.65 | 24.28 | 100 |
| 1.4 | 79.85 | 89.88 | 32.99 | 100 |

Table 13: Effects of the hyper-parameter $\lambda_4$.

| $\lambda_4$ | BA (%) | WSR (%) | Chamfer | PLG (%) |
|---|---|---|---|---|
| 0.6 | 74.57 | 92.56 | 24.68 | 100 |
| 0.8 | 81.16 | 87.71 | 26.37 | 100 |
| 1.0 | 78.33 | 93.68 | 23.04 | 100 |
| 1.2 | 75.90 | 88.05 | 24.03 | 100 |
| 1.4 | 80.74 | 87.30 | 24.02 | 100 |

Table 14: Effects of the hyper-parameter $\mu_0$.

| $\mu_0$ | BA (%) | WSR (%) | Chamfer | PLG (%) |
|---|---|---|---|---|
| 0.0006 | 84.17 | 87.53 | 21.50 | 100 |
| 0.0008 | 81.18 | 89.70 | 23.47 | 100 |
| 0.0010 | 78.33 | 93.68 | 23.04 | 100 |
| 0.0012 | 82.78 | 90.55 | 17.46 | 100 |
| 0.0014 | 81.76 | 84.80 | 29.49 | 100 |

## H    ADDITIONAL RESULTS OF THE EFFECTS OF KEY HYPER-PARAMETERS

In this section, similar to the main manuscript, we adopt the CIFAR-10 dataset as an example to discuss the effects of other key hyper-parameters involved in our GLBW.

### H.1    EFFECTS OF THE HYPER-PARAMETER $\lambda_3$

As shown in Table 12, the hyper-parameter $\lambda_3$ has a mild influence on our GLBW, especially on its trigger generalization. Specifically, our method can reach a 100% PLG under all settings. In other words, users can achieve faithful SRV evaluation based on our GLBW without fine-tuning $\lambda_3$.

### H.2    EFFECTS OF THE HYPER-PARAMETER $\lambda_4$

As shown in Table 13, the hyper-parameter $\lambda_4$ also has a mild influence on our GLBW, although it has higher impacts on WSR. This is mostly because $\lambda_4$ relates to the 'suppression' of potential triggers that may have adverse effects on the WSR. Nevertheless, our method can reach a 100% PLG under all settings. Accordingly, users can achieve faithful SRV evaluation based on our GLBW without fine-tuning this hyper-parameter.

### H.3    EFFECTS OF THE HYPER-PARAMETER $\mu_0$

As shown in Table 14, $\mu_0$ also has mild effects to trigger the generalization of our GLBW. It is mostly because of the adaptive optimization process of generating synthesized triggers. Users can still achieve faithful SRV evaluation based on our GLBW without fine-tuning this hyper-parameter.

## I    A CLOSER LOOK TO THE EFFECTIVENESS OF OUR GLBW

In this section, we further explore why our GLBW can reduce trigger generalization by visualizing all synthesized triggers via principal component analysis (PCA).

**Settings.** We use the vanilla, BWTP and GLBW models trained in Section 5.2 for experiments. Similar to (Qiao et al., 2019), for each model, we sample 1,000 triggers produced by neural cleanse and embed these triggers into a 2-D space via PCA to visualize trigger distribution. In particular, we train PCA on the triggers generated from the vanilla model, and then apply it to the triggers generated from all the models. That is to say, all the dimension reduction processes are under the same linear transformation.

**Results.** As shown in Figure 15, the effective triggers generated from the model watermarked by our GLBW are clustered together in all cases. In other words, they are from a similar distribution.Accordingly, our GLBW has a low trigger generalization. In contrast, the effective triggers generated from the model watermarked by vanilla backdoor attack scatter in the whole space, indicating that they are from different latent distributions. As such, the vanilla backdoor watermark

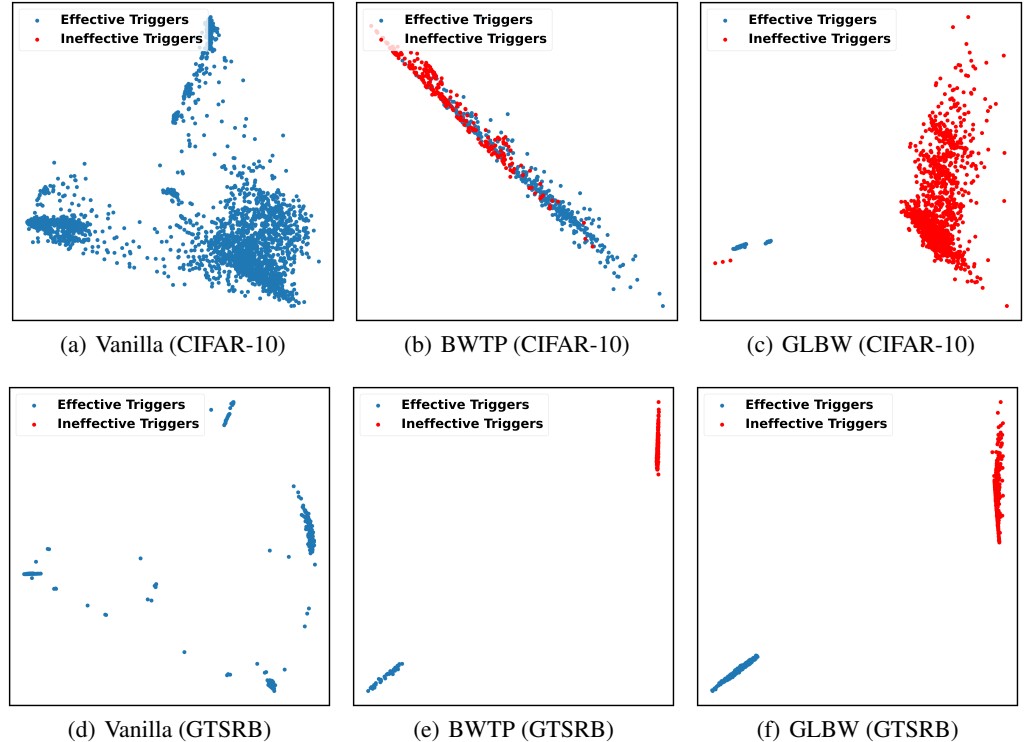

Figure 15: The visualization of the trigger distribution.

has a high trigger generalization. The effective triggers of BWTP have similar behavior to that of GLBW on GTSRB while having a similar behavior to the vanilla backdoor watermark on CIFAR-10. These phenomena are consistent with the trigger generalization of BWTP (low generalization on the GTSRB dataset and high generalization on the CIFAR-10 dataset).

## J  WHY WOULD OUR METHODS DECREASE BENIGN ACCURACY?

In our methods, we treat the universal adversarial perturbations (UAPs) and potential triggers the same, whereas we have to admit that they may be different. Accordingly, our methods (especially GLBW) are similar to adversarial training against UAPs in terms of effectiveness. As such, our GLBW has a similar effect as conducting adversarial training on UAPs that will (significantly) decreases benign accuracy especially when the task is relatively complicated.

However, it does not necessarily means that our method is not practical even we treat them the same.

Firstly, having a relatively low benign accuracy will not reduce the reliability and practicality of our method. Firstly, our evaluation is based on poisoned samples instead of clean samples. As such, we only need to ensure that our methods lead to a high watermark success rate (instead of a high benign accuracy) for faithful results. The watermark success rates are higher than 90% in all cases, which is sufficiently high; Secondly, as we mentioned in our experiments, the watermarked DNNs are only used for evaluating SRV methods instead of for deployment. Accordingly, the decrease in clean accuracy led by our method will not hinder its usefulness.

Secondly, our method can still reduce trigger generalization since it minimizes both trigger generalization and the risk of UAPs simultaneously, even though we treat them the same.

Thirdly, minimizing the risk of UAPs may have potential benefits in reducing trigger generalization. (Ilyas et al., 2019) revealed that adversarially robust models focus more on 'robust features' instead of non-robust ones (*e.g.*, textures). Accordingly, our method may makes DNNs rely more on the original trigger pattern for poisoned samples and therefore could reduce trigger generalization.

Table 15: The average rank of SRV methods that is evaluated with our GLBW method and human inspection on the GTSRB dataset.

| Dataset↓ | SRV→ Method↓ | BP | GBP | GCAM | GGCAM | OCC | FA | LIME |
|---|---|---|---|---|---|---|---|---|
| GTSRB | Ours | 4.00 | 4.00 | 6.00 | 6.00 | 2.00 | 3.00 | 1.00 |
| | Human | 4.00 | 4.00 | 6.00 | 6.00 | 2.00 | 3.00 | 1.00 |



(a) $4 \times 4$ square      (b) $5 \times 5$ square      (c) Pencil      (d) Triangle

Figure 16: Four additional trigger patterns used in our evaluation.

Table 16: The IOU of SRV methods with the standardized XAI evaluation on the CIFAR-10 dataset and the GTSRB dataset. **Square**: the trigger with the square-like pattern; **Pencil**: the trigger with the pencil-like pattern; **Triangle**: the trigger with the triangle-like pattern. All trigger patterns have the same size (*i.e.*, 9 white pixels).

| Dataset↓ | SRV→ Trigger↓ | BP | GBP | GCAM | GGCAM | OCC | FA | LIME |
|---|---|---|---|---|---|---|---|---|
| | Square | 0.2123 | 0.2123 | 0.0000 | 0.2068 | 0.8849 | 0.4935 | 0.9792 |
| CIFAR-10 | Pencil | 0.1675 | 0.1675 | 0.0000 | 0.1653 | 0.6776 | 0.7228 | 0.9956 |
| | Triangle | 0.2534 | 0.2534 | 0.0000 | 0.2489 | 0.6777 | 0.8463 | 0.9771 |
| | Square | 0.3438 | 0.3438 | 0.0000 | 0.3151 | 0.7388 | 0.4233 | 1.0000 |
| GTSRB | Pencil | 0.2604 | 0.2604 | 0.0000 | 0.2225 | 0.8978 | 0.7570 | 0.9979 |
| | Triangle | 0.3003 | 0.3003 | 0.0000 | 0.2762 | 0.5627 | 0.4371 | 0.9890 |

Lastly, UAPs can be regarded as the trigger of the 'natural backdoor' of models learned from samples (Wenger et al., 2022). They have very similar properties to backdoor triggers and therefore it is very difficult (or probably even impossible) to distinguish them from potential triggers.

## K    ADDITIONAL EXPERIMENTS OF HUMAN INSPECTION

To further verify that our GLBW-based XAI evaluation is faithful, we generate 10 groups of saliency maps based on 10 randomly selected images on the GTSRB dataset. We conduct human inspection experiments by inviting 10 people and asking them to grade all groups of saliency maps independently. As shown in Table 15, the rankings generated by our GLBW method are similar to those made by people. These results verify that an SRV method with a higher order in the ranking list generated by our method is truly better than those with lower orders.

## L    ADDITIONAL EXPERIMENTS OF STANDARDIZED XAI EVALUATION WITH DIFFERENT TRIGGERS

### L.1    RESULTS WITH DIFFERENT TRIGGER SHAPES

To explore whether the trigger shape has significant effects to our standardized XAI evaluation, we evaluate the SRV methods with different trigger shapes, including square, pencil, and triangle (as shown in Figure 16), on CIFAR-10 and GTSRB datasets. In this experiment, all evaluated trigger

Table 17: The rank of SRV methods with the standardized XAI evaluation on the CIFAR-10 dataset and the GTSRB dataset. **Square**: the trigger with the square-like pattern; **Pencil**: the trigger with the pencil-like pattern; **Triangle**: the trigger with the triangle-like pattern. All trigger patterns have the same size (*i.e.*, 9 white pixels).

| Dataset↓ | SRV→ Trigger↓ | BP | GBP | GCAM | GGCAM | OCC | FA | LIME |
|---|---|---|---|---|---|---|---|---|
| CIFAR-10 | Square | 4 | 4 | 7 | 6 | 2 | 3 | 1 |
| | Pencil | 4 | 4 | 7 | 6 | 3 | 2 | 1 |
| | Triangle | 4 | 4 | 7 | 6 | 3 | 2 | 1 |
| GTSRB | Square | 4 | 4 | 7 | 6 | 2 | 3 | 1 |
| | Pencil | 4 | 4 | 7 | 6 | 2 | 3 | 1 |
| | Triangle | 4 | 4 | 7 | 6 | 2 | 3 | 1 |

Table 18: The IOU of SRV methods with the standardized XAI evaluation on the CIFAR-10 dataset and the GTSRB dataset. $3 \times 3$: the trigger with the $3 \times 3$ square-like pattern; $4 \times 4$: the trigger with the $4 \times 4$ square-like pattern; $5 \times 5$: the trigger with the $5 \times 5$ square-like pattern.

| Dataset↓ | SRV→ Size↓ | BP | GBP | GCAM | GGCAM | OCC | FA | LIME |
|---|---|---|---|---|---|---|---|---|
| CIFAR-10 | $3 \times 3$ | 0.2123 | 0.2123 | 0.0000 | 0.2068 | 0.8849 | 0.4935 | 0.9792 |
| | $4 \times 4$ | 0.2867 | 0.2867 | 0.0000 | 0.2814 | 0.8796 | 0.9821 | 0.9841 |
| | $5 \times 5$ | 0.3811 | 0.3811 | 0.0000 | 0.3798 | 0.7255 | 0.6672 | 0.9923 |
| GTSRB | $3 \times 3$ | 0.3438 | 0.3438 | 0.0000 | 0.3151 | 0.7388 | 0.4233 | 1.0000 |
| | $4 \times 4$ | 0.3360 | 0.3360 | 0.0000 | 0.2857 | 0.7803 | 1.0000 | 0.9914 |
| | $5 \times 5$ | 0.3436 | 0.3436 | 0.0000 | 0.2861 | 0.8364 | 0.5205 | 0.9838 |

Table 19: The rank of SRV methods with the standardized XAI evaluation on the CIFAR-10 dataset and the GTSRB dataset. $3 \times 3$: the trigger with the $3 \times 3$ square-like pattern; $4 \times 4$: the trigger with the $4 \times 4$ square-like pattern; $5 \times 5$: the trigger with the $5 \times 5$ square-like pattern.

| Dataset↓ | SRV→ Size↓ | BP | GBP | GCAM | GGCAM | OCC | FA | LIME |
|---|---|---|---|---|---|---|---|---|
| CIFAR-10 | $3 \times 3$ | 4 | 4 | 7 | 6 | 2 | 3 | 1 |
| | $4 \times 4$ | 4 | 4 | 7 | 6 | 3 | 2 | 1 |
| | $5 \times 5$ | 4 | 4 | 7 | 6 | 2 | 3 | 1 |
| GTSRB | $3 \times 3$ | 4 | 4 | 7 | 6 | 2 | 3 | 1 |
| | $4 \times 4$ | 4 | 4 | 7 | 6 | 3 | 1 | 2 |
| | $5 \times 5$ | 4 | 4 | 7 | 6 | 2 | 3 | 1 |

patterns have 9 pixels. As shown in Table 16-17, the trigger shape has mild effects to the final ranking, although it may have some influences to IOU values.

## L.2 Results with Different Trigger Sizes

To explore whether the trigger size has significant effects to our standardized XAI evaluation, we evaluate SRV methods based on square-type triggers with different sizes (*i.e.*, $3 \times 3$, $4 \times 4$ and $5 \times 5$, as shown in Figure 16), on CIFAR-10 and GTSRB datasets. As shown in Table 18-19, the trigger shape has minor effects to the final ranking, although it may have some influences to IOU values.

## M  Connections and Differences with Related Works

In this section, we discuss the connections and differences between our methods (*i.e.*, BWTP and GLBW) and backdoor attacks, model watermarks, adversarial learning, and neural cleanse.

## M.1 Connections and Differences with Backdoor Attacks

In general, both our methods and backdoor attacks intend to implant special model behaviors (*i.e.*, backdoor) to DNNs. However, they still have many intrinsic differences.

Firstly, backdoor attacks aim at maliciously manipulating attacked models during the inference process. In contrast, our watermarks are designed for the evaluation of saliency-based representation visualization (SRV), which is a positive application.

Secondly, there are different parties (*i.e.*, adversaries, and victim model users) involved in backdoor attacks while only the adjudicator is included in our SRV evaluation methods. Accordingly, backdoor attacks need to ensure that the attacked model has high benign accuracy or its poisoned samples are similar to their benign version to circumvent the detection of victim model users. However, in our methods, we only need to ensure a high watermark/attack success rate (instead of the high benign accuracy) of watermarked DNNs since we don't need to fulfill watermark stealthiness.

Lastly, backdoor attacks only need to ensure that the original trigger pattern used for training can activate hidden backdoors in attacked DNNs. In other words, the backdoor adversaries don't need to measure or control the degree of trigger generalization. However, as we mentioned in the main manuscript, our methods need to minimize the generalization to design faithful SRV evaluation.

## M.2 Connections and Differences with Model Watermarks

Currently, there are different types of model watermarks. Among all watermarking techniques, backdoor-based methods are probably the most classical and widespread ones. In general, our methods share some similarities to backdoor-based model watermarks. For example, both of them assume that the defenders can fully control the training process of models and intend to implant backdoors to watermarked DNNs. However, they still have some essential differences.

Firstly, similar to backdoor attacks, there are different parties (*i.e.*, the model owner and users/adversaries) involved in model watermarks. Accordingly, the model owner still requires to preserve the benign accuracy of watermarked DNNs otherwise users will not exploit them. In contrast, it is not necessary to have a high benign accuracy in our methods.

Secondly, model watermarks need to ensure robustness under different watermark-removal attacks (*e.g.*, fine-tuning, and model pruning) since adversaries have the incentive to remove them. However, in our methods, we only require to have a low trigger generalization of backdoor watermarks.

## M.3 Connections and Differences with Adversarial Learning

Both (targeted universal) adversarial attacks and our methods (during the training process) intend to find a pattern that can mislead model predictions to a specific class. Instead of simply having a high attack effectiveness as required by adversarial attacks, our methods also need to ensure that the generated perturbation is (significantly) different from the original trigger pattern. This requirement is necessary for minimizing trigger generalization of backdoor watermarks.

Both our methods and adversarial training formulate model training as a min-max problem. However, adversarial training aims to make models robust to all potential perturbations. In contrast, our watermarks intend to preserve the backdoor regarding the original trigger while reducing other backdoors instead of eliminating all backdoors.

## M.4 Connections and Differences with Neural Cleanse

In general, both our methods and neural cleanse (Wang et al., 2019) intend to generate targeted universal adversarial perturbations as potential trigger patterns. However, our methods have different constraints (*e.g.*, $|\boldsymbol{m}' \cap \boldsymbol{m}| \leq \tau \cdot |\boldsymbol{m}|$) since we have fundamentally different purposes.

Firstly, neural cleanse has no information (*e.g.*, trigger size and target label) about the ground-truth trigger pattern. Accordingly, it needs to generate potential trigger towards each class and adopt trigger sparsity to decide the final one. However, its assumption that all trigger patterns should be sparse is not hold for advanced backdoor attacks. As such, this method is not effective (Li et al., 2021b). In contrast, we know the ground-truth trigger patch, its target label, and its sparsity. Our

Table 20: The rank (based on average IOU among all poisoned sample) across four trigger patterns of SRV methods that is evaluated with different methods.

| Dataset | Method, SRV | BP | GBP | GCAM | GGCAM | OCC | FA | LIME |
|---------|-------------|------|------|------|-------|------|------|------|
| CIFAR-10 | Ours | 4.75 | 4.75 | 7 | 4.5 | 2 | 3 | 1 |
| | Adaptive | 4.4 | 4.5 | 7 | 5 | 2.5 | 2.5 | 1 |
| GTSRB | Ours | 4.5 | 4.5 | 7 | 5 | 2 | 2.5 | 1.25 |
| | Adaptive | 4.75 | 4.75 | 6.5 | 5 | 2.25 | 2.75 | 1 |

purpose is to find potential triggers with the highest attack effectiveness and differences from the ground-truth one instead of finding the ground-truth trigger.

Secondly, neural cleanse exploited the standard SGD to solve the optimization problem. In contrast, in our GLBW, we design an adaptive optimization method to find desired potential trigger patterns since our task is significantly more difficult than that of the neural cleanse. Please refer to Appendix E for more details about our optimization process.

## N ADDITIONAL EXPERIMENTS OF STANDARDIZED EVALUATION METHOD WITH ADAPTIVE BINARIZATION

Regarding the selection of $M$ pixels proposed in our standardized backdoor-based XAI evaluation, people may argue that it is not significantly better than the previous method. In this section, we discuss whether it would be better if we use adaptive binarization methods to obtain regions with high influences to calculate the rank of each SRV method.

**Settings.** We hereby exploit a classical adaptive binarization method, *i.e.*, OTSU (Otsu, 1979), to obtain regions with high influences to calculate the rank of each SRV method and compare the results of our standardized method (dubbed 'Ours') and those of the standardized method with adaptive binarization (dubbed 'Adaptive'). All other settings are the same as those used in Section 3.1.

**Results.** As shown in Table 20, using adaptive binarization method has a mild influence to the final average ranking. These results verify the effectiveness of our method.

## O REPRODUCIBILITY STATEMENT

In the appendix, we provide detailed descriptions of the datasets, models, training and evaluation settings, and computational facilities. The codes and model checkpoints for reproducing the main experiments of our evaluation are also provided in the supplementary material. Our codes are available at `https://github.com/yamengxi/GLBW`.

## P DISCUSSIONS ABOUT ADOPTED DATA

In this paper, all adopted samples are from the open-sourced datasets (*i.e.*, CIFAR-10 and GTSRB). These datasets contain no human object. We admit that we modified a few samples for watermarking. However, our research treats all samples the same and all triggers contain no offensive content. Accordingly, our work fulfills the requirements of these datasets and has no privacy violation.

