# OpenReview forum: "Towards Faithful XAI Evaluation via Generalization-Limited Backdoor Watermark"
_ICLR.cc/2024/Conference — ICLR 2024 poster_

### Official Review · Reviewer_RMSQ · 2023-10-27

**Soundness:** 3 good
**Presentation:** 3 good
**Contribution:** 3 good
**Rating:** 8
**Confidence:** 2

**Summary:**

The paper revisits the evaluation techniques for saliency-based representation visualization (SRV). The authors uncover the unreliable nature of current backdoor-based SRV evaluation methods, particularly due to the trigger generalization of backdoor watermarks. They also highlight implementation limitations. From these insights, they introduce a generalization-limited backdoor watermark and, based on this, design a more accurate XAI evaluation. The aim of their research is to enhance the understanding of XAI evaluation and aid in the creation of more interpretable deep learning methods.

**Strengths:**

1. The paper highlights a crucial insight into the shortcomings of the current backdoor-based SRV evaluation and presents a new method called GLBW for a more accurate XAI assessment.
2. The authors have conducted extensive experiments on benchmark datasets to verify the effectiveness of their GLBW, adding credibility to their claims.
3. The authors have provided detailed descriptions of datasets, models, training, and evaluation settings in the appendix. They also commit to releasing training codes upon acceptance, promoting transparency and reproducibility.

**Weaknesses:**

The only weakness I consider is that the authors haven't provided enough empirical evidence demonstrating the superiority of their proposed method over traditional backdoor-based techniques. For instance, they don't experimentally address why the minimum bounding box approach might yield inaccurate results as they claim.

**Questions:**

Please refer to the weaknesses.

---

> ### Author Response · Authors · 2023-11-19
> **Author Response**
>
> Dear Reviewer RMSQ, we sincerely thank you for your valuable time and comments. We are encouraged by your positive comments on **insightful analyses**, **extensive experiments**, **good method**, **good contribution**, and **good writing**. We hope the following responses could help clarify potential misunderstandings and alleviate your concerns.
>
> ---
> **Q1**: The only weakness I consider is that the authors haven't provided enough empirical evidence demonstrating the superiority of their proposed method over traditional backdoor-based techniques. For instance, they don't experimentally address why the minimum bounding box approach might yield inaccurate results as they claim.
>
> **R1**: Thank you for this constructive comment! We are deeply sorry that our previous submission failed to explain it more clearly.
>
> - **Why taking absolute values of gradients in all (instead of parts of) evaluated SRV methods is better**: Given an image and a trained model, existing SRV methods need to obtain the 'influence value' (e.g., gradient) of each pixel to model's prediction of the image before generating its saliency map. **The influence value can be positive or negative**. Its positive value indicates that increasing its pixel value will have a positive effect on the prediction. In general, **to obtain the saliency map, we usually need to take the absolute value of each influence value before SRV methods filter out the most critical pixel positions** (with sufficiently high influence scores). Otherwise, pixels having significantly negative effects on the prediction will not be selected. However, the existing backdoor-based SRV evaluation **[13] only took the absolute value for BP while keeping the original influence value for other SRV methods** (i.e., GBP, GCAM, GGCAM, OCC, FA, LIME). **This inconsistent setting will lead to unreliable results**. Accordingly, our standardized method is better than the traditional one.
> - **Why selecting $M$ pixel locations with the maximum saliency value (instead of using pre-defined threshold) as significant regions is better**: The same threshold value may have significantly different influences across samples, even for the same SRV method, especially on different datasets. Accordingly, **the saliency areas generated by the traditional method are not stable and reliable**. However, **using our TOP-K selection method can mitigate this problem** and therefore it is better than the traditional method.
> - **Why calculating the IOU based on the significant regions directly is better than using minimum bounding box**: Using the minimum bounding box may lead to a false sense of high performance. For example, consider a two-pixel trigger pattern where one pixel is located in the upper left corner and the other in the lower right corner. Assume that the saliency map is one pixel in the lower left corner and one pixel in the upper right corner. In this case, if the minimum bounding box is used, then IoU=1. However, this saliency map is completely wrong since there is no overlapping region. In other words, **using traditional methods may result in IoU values that cannot correctly reflect the substantive effects of XAI methods**. In contrast, using our method can avoid this problem. Accordingly, it is better than the traditional method.
> - To further alleviate your concerns, we have also compared the results of the traditional method and those of our method. As shown in the following Table 1, the vanilla (i.e., traditional) method has various evaluations of the performance of the same SRV method on different datasets (especially for BP). In contrast, our standardized evaluation method has consistent rankings. These results verify that our method is more faithful.
>
>
> **Table 1.** The average rank (based on the IOU) of SRV methods that is evaluated with the vanilla backdoor-based method and its standardized version on CIFAR-10 and GTSRB datasets
> | Method$\downarrow$ | Dataset$\downarrow$, SRV$\rightarrow$ | BP | GBP | GCAM | GGCAM | OCC | FA | LIME |
> |:------------:|:--------:|:---:|:---:|:---:|:---:|:---:|:---:|:---:|
> | Vanilla      | CIFAR-10 |5.25 |3.00 |6.25 |3.25 |1.50 |2.50 |6.25 |
> | Vanilla      | GTSRB    |2.75 |3.75 |6.50 |3.75 |2.50 |2.25 |6.50 |
> | Standardized | CIFAR-10 |4.75 |4.75 |7.00 |4.50 |2.00 |3.00 |1.00 |
> | Standardized | GTSRB    |4.50 |4.50 |7.00 |5.00 |2.00 |2.75 |1.25 |
>
>
> ---

---

> ### Author Response · Authors · 2023-11-21
> **Thanks to Reviewer RMSQ**
>
> Please allow us to thank you again for reviewing our paper and the valuable feedback, and in particular for recognizing the strengths of our paper in terms of *insightful analyses*, *extensive experiments*, *good method*, *good contribution*, and *good writing*.
>
> Kindly let us know if our response and the new experiments have properly addressed your concerns. We are more than happy to answer any additional questions during the post-rebuttal period. Your feedback will be greatly appreciated.

---

> ### Author Response · Authors · 2023-11-22
> **A Gentle Reminder of the Final Feedback**
>
> We would like to thank the reviewer for the helpful discussion during the first round of the review. We hope our response has adequately addressed your concerns. We take this as a great opportunity to improve our work and shall be grateful for any additional feedback you could give to us.

---

> ### Author Response · Authors · 2023-11-23
> **A Second Reminder of the Post-rebuttal Feedback**
>
> Dear Reviewer RMSQ,
>
> We greatly appreciate your initial comments. We totally understand that you may be extremely busy at this time. But we still hope that you could have a quick look at our responses to your concerns. We appreciate any feedback you could give to us. We also hope that you could kindly update the rating if your questions have been addressed. We are also happy to answer any additional questions before the rebuttal ends.
>
> Best Regards,
>
> Paper3310 Authors

---

### Official Review · Reviewer_ZYFJ · 2023-10-31

**Soundness:** 3 good
**Presentation:** 4 excellent
**Contribution:** 4 excellent
**Rating:** 8
**Confidence:** 4

**Summary:**

This paper explores how to faithfully evaluate XAI methods based on backdoor watermarks. Specifically, the authors first reveal the implementation limitations and unreliable nature of using standard backdoor watermarks due to trigger generalization. Based on their analyses, the authors propose the first generalization-limited backdoor watermark to further design more faithful XAI evaluation. The authors evaluate their method on CIFAR-10 and GTSRB datasets.

**Strengths:**

1.	This work cleverly links two seemingly unrelated (i.e., backdoor attacks and model interpretability) yet important research fields. In particular, the proposed method circumvents the harmful nature of backdoor attacks since the watermarked model is only used for evaluating models instead of for deployment. Although this perspective is first proposed in a previous work, further non-trivial exploration of this angle is also very meaningful and valuable.
2.	I enjoy the analyses of the limitations of existing backdoor-based XAI evaluations, especially their unreliable nature. These findings are non-trivial and are critical for practical backdoor-based XAI evaluations.
3.	The proposed method is novel and reasonable. I think the authors have comprehensively demonstrate the design philosophy of their methods.
4.	The experiments are comprehensive to a large extent. In particular, the authors exploit different trigger inversion techniques to evaluate trigger generalization, which should be encouraged.
5.	The paper is well-written and its main idea is easy to follow.

**Weaknesses:**

1.	It would be better if the authors can provide more details about the differences between GLBW and BWTP. They seem similar in terms of formulas alone.
2.	Regarding the selection of M pixels, I do not think it is significantly better than the previous method. One example is that the size of the calculated saliency is significantly larger than M, while the size of the trigger is small. Because of M, the IoU value is high, but in reality, it is small due to the union.
3.	The paper treats the universal adversarial perturbations and the triggers the same, while their characteristics differ. However, the proposed method treats them the same. Please provide more explanations about why it is not a problem or how to address it.
4.	It would be better if the author could provide more details and discussions about why trigger generalization is very important for XAI evaluation in the appendix.
5.	The explainability metric is based on simple backdoored patterns, but why would it be a good reference for complicated features in real images for practice?

**Questions:**

See the above weakness.

---

> ### Author Response · Authors · 2023-11-19
> **Author Response (Part I)**
>
> Dear Reviewer ZYFJ, we sincerely thank you for your valuable time and comments. We are encouraged by your positive comments on our **meaningful and valuable exploration**, **non-trivial findings**, **novel and reasonable methods**, **comprehensive experiments**, and **good paper writing**. We hope the following responses could help clarify potential misunderstandings and alleviate your concerns.
>
> ---
>
> **Q1**: It would be better if the authors can provide more details about the differences between GLBW and BWTP. They seem similar in terms of formulas alone.
>
> **R1**: Thanks for the insightful comment! We are deeply sorry that our submission may lead you to some misunderstandings that we want to clarify.
>
> - **Eq.(2) is different from Eq.(1)**. Formally, penalty loss (contained in Eq.(1)) and generalization loss (contained in Eq.(2)) have fundamental differences. Firstly, penalty loss is a maximization while generalization loss is essentially a minimization since its objective is $-\mathcal{L}$ instead of $\mathcal{L}$; Secondly, the constraint in penalty loss is $|m'|\leq t$ while that in generalization loss is $|m' \cap m| \leq \tau \cdot |m|$; Thirdly, there is a term $\mu \cdot |m'|$ in generalization loss that is not included in penalty loss.
> - **Penalty loss and generalization loss have fundamentally different meanings**. In general, **penalty loss synthesizes the potential trigger pattern and penalizes its effects based on its distance to the original one**. If the potential trigger pattern is close to the ground-truth one, the predictions of watermarked DNNs to samples containing this pattern should be similar to the target label; Otherwise, their predictions should be similar to their ground-truth label. In contrast, **generalization loss generates the most effective potential trigger pattern (other than the original one) and then minimizes its effects**.
> - **Eq.(2) is feasible to reduce trigger generalization since it can generate potential trigger pattern (other than the original one) and minimizes its effects**. In particular, as we mentioned in Section 4.3 and illustrated in Appendix (Section 4), **we designed an adaptive optimization method to find promising synthesized patterns** with the highest attack effectiveness and differences from the original trigger. It is also important for the success of our GLBW method.
>
> We have added more details in Section 4.2 and Section 4.3 to better clarify Eq.(1) \& Eq.(2) in our revision to avoid potential misunderstandings.
>
> ---

---

> > ### Author Response · Authors · 2023-11-19
> > **Author Response (Part II)**
> >
> > ---
> >
> > **Q2**: Regarding the selection of M pixels, I do not think it is significantly better than the previous method. One example is that the size of the calculated saliency is significantly larger than M, while the size of the trigger is small. Because of M, the IoU value is high, but in reality, it is small due to the union.
> >
> > **R2**: Thank you for this insightful comment! We hereby provide more explanations to alleviate your concerns.
> >
> > - Existing SRV methods require setting a pre-defined threshold to filter out the most critical regions. However, due to different factors (e.g., the way of calculating influence scores), different SRV methods have very different values even under the same setting. Accordingly, **setting the same threshold to compare them is unfair**. Besides, **it is hard to select an 'optimal threshold' for each method without human inspection**.
> > - We argue that selecting $M$ pixel locations with the maximum saliency value as significant regions for analysis is a feasible solution to a large extent. Specifically, they can be regarded as the most critical regions concentrated by the evaluated SRV method, although there might be larger significant areas with relatively high saliency value as you mentioned. **Comparing the ability of different SRV methods to recognize the most critical regions of the same size is a more equitable approach**.
> > - However, we do understand your concerns. We hereby exploit a classical adaptive binarization method (i.e., [OTSU](https://cw.fel.cvut.cz/b201/_media/courses/a6m33bio/otsu.pdf)) to obtain regions with high influences to calculate the rank of each SRV method and compare the results of our standardized method (dubbed 'Ours') and those of standardized method with adaptive binarization (dubbed 'Adaptive'). As shown in the following table, **using adaptive binarization method has a mild influence to the final average ranking**. These results verify the effectiveness of our method.
> >
> > **Table 1.** The rank (based on average IOU among all poisoned sample) across four trigger patterns of SRV methods that is evaluated with different methods.
> >
> > |  Dataset$\downarrow$ | Method$\downarrow$, SRV$\rightarrow$ |  BP  |  GBP | GCAM | GGCAM |  OCC |  FA  | LIME |
> > |:--------:|:---------:|:----:|:----:|:----:|:-----:|:----:|:----:|:----:|
> > | CIFAR-10 |    Ours   | 4.75 | 4.75 |   7  |  4.5  |   2  |   3  |   1  |
> > | CIFAR-10 |  Adaptive |  4.5 |  4.5 |   7  |   5   |  2.5 |  2.5 |   1  |
> > |   GTSRB  |    Ours   |  4.5 |  4.5 |   7  |   5   |   2  | 2.75 | 1.25 |
> > |   GTSRB  |  Adaptive | 4.75 | 4.75 |  6.5 |   5   | 2.25 | 2.75 |   1  |
> >
> >
> > We haved provided more details in Appendix N of our revision. Besides, due to the limitation of rebuttal time, we are not able to re-calculate all results with the adaptive binarization method. However, if you think it is necessary, we promise that we will conduct the remaining experiments after the rebuttal and provide all of them in the appendix of our final version.
> >
> >
> > ---
> >
> > **Q3**: The paper treats the universal adversarial perturbations and the triggers the same, while their characteristics differ. However, the proposed method treats them the same. Please provide more explanations about why it is not a problem or how to address it.
> >
> > **R3**: Thank you for this insightful comment! We hereby provide more explanations to alleviate your concerns.
> >
> > - We admit that we treat universal adversarial perturbations (UAPs) and triggers equally in our optimization process designed for reducing trigger generalization, as you suggested. In general, universal adversarial perturbations can be regarded as the trigger of the ‘natural backdoor’ of models learned from samples ([Wenger et al. 2022](https://proceedings.neurips.cc/paper_files/paper/2022/file/8af749935131cc8ea5dae4f6d8cdb304-Paper-Datasets_and_Benchmarks.pdf)). Accordingly, they have very similar properties to backdoor triggers and therefore **it is very difficult (or probably even impossible) to distinguish them from original and potential triggers**.
> > - **Even though we treat them the same, our methods can still reduce trigger generalization** since they minimize both trigger generalization and the risk of UAPs simultaneously during their optimization process.
> > - Besides, **minimizing the risk of UAPs may have potential benefits in reducing trigger generalization**. Recent studies (e.g., [Andrew et al. 2019](https://proceedings.neurips.cc/paper_files/paper/2019/file/e2c420d928d4bf8ce0ff2ec19b371514-Paper.pdf)) revealed that adversarially robust models focus more on 'robust features' instead of non-robust ones (e.g., textures). Accordingly, our method may makes DNNs rely more on the original trigger pattern for poisoned samples and therefore could reduce trigger generalization.
> >
> > We have provided more discussions in Appendix J of our revision.
> >
> > ---

---

> > > ### Author Response · Authors · 2023-11-19
> > > **Author Response (Part III)**
> > >
> > > ---
> > >
> > > **Q4**: It would be better if the author could provide more details and discussions about why trigger generalization is very important for XAI evaluation in the appendix.
> > >
> > > **R4**: Thank you for this quesion and we are deeply sorry that we failed to explain it more clearly in our submission. In general, **position generalization will make the result of backdoor-based SRV evaluation less reliable**. More details are as follows:
> > > - **Backdoor-based SRV evaluation methods rely on a latent assumption that only the trigger used for training (dubbed 'original trigger') can activate backdoors**. These methods believe that trigger regions should be treated as the regions that contribute the most to the model's prediction (i.e., target label) of poisoned samples because their ground-truth labels are not the target label. Accordingly, they use the area of original trigger as the ground-truth reference and calculate the average intersection over union (IOU) between it and the saliency areas generated by the SRV method of the backdoored model over different backdoored samples as an indicator to evaluate the SRV method.
> > > - **This assumption does not hold when backdoor watermark has position generalization**. As we shown in our Figure 4, there are many potential trigger patterns other than the original one that can still activate backdoors. In other words, **this assumption does not hold for existing watermarks**.
> > > - **Its failure may lead to unreliable results**. For example, given a SRV method, assume that its generated saliency areas of most poisoned samples are only a small part of that of the original trigger. According to backdoor-based SRV evaluation approaches, this SRV method will be treated very poorly since it has a small IOU. However, due to the generalization of backdoor watermarks, the model may learn this local region rather than the whole trigger. In this case, the evaluated SRV method is in fact highly effective, contradicting to the results of existing backdoor-based SRV evaluation methods.
> > >
> > > We have provided more details in the introduction and Appendix C of our revision to make it more clearly.
> > >
> > >
> > > ---
> > >
> > >
> > > **Q5**: The explainability metric is based on simple backdoored patterns, but why would it be a good reference for complicated features in real images for practice?
> > >
> > > **R5**: Thank you for this insightful question! We admit that our GLBW and other existing backdoor-based SRV methods all rank SRV methods based on their performance in explaining simple backdoored patterns (instead of high-level complicated features). However, **it not necessarily means that our method is impractical**. We hereby provide more explanations:
> > >
> > > - In practice, **humans categorize a given image based on its local regions in most cases**. For example, we only need to see the image areas of its 'head' to know it is a bird. **These local regions are similar to the trigger patterns** (e.g., a patch) used in our method. Accordingly, results made by our method can be a good reference in real images for practice.
> > > - Currently, **it is impossible to faithfully evaluate the performance of SRV methods for complicated features** since there is no ground-truth salience map for them. Even a human expert cannot mark the salience map for complicated features.
> > > - Simple backdoored patterns are also features used by DNNs for their predictions. Accordingly, **the evaluation of SRV methods on trigger features is the first and the most important step toward evaluating their general performance** and is therefore of great significance.
> > >
> > > We have provided more details in Appendix C of our revision to make it more clearly.
> > >
> > > ---

---

> > > > ### Comment · Reviewer_ZYFJ · 2023-11-21
> > > >
> > > > After reading the rebuttal, my concerns have been well solved. Considering the novelty and solid experiments, I tend to increase my original score to "Accept".

---

> > > > > ### Author Response · Authors · 2023-11-21
> > > > > **Thank You for Your Positive Feedback!**
> > > > >
> > > > > Thank you so much for your positive feedback! It encourages us a lot.

---

### Official Review · Reviewer_4nxh · 2023-10-31

**Soundness:** 2 fair
**Presentation:** 3 good
**Contribution:** 3 good
**Rating:** 6
**Confidence:** 3

**Summary:**

In this paper, the authors aim at the evaluation problem of saliency-based representation visualization (SRV). For that, they firstly argue that the current backdoor-based SRV evaluation methods have the implementation limitations and unreliable nature. Based on that, they propose the faithful XAI method to overcome these issues. A variety of experiments verify the effectiveness of proposed method.

**Strengths:**

1. The writing or story is good, they provide a solid experiment to show the issues of current backdoor-based SRV evaluation, and then give a reasonable solution to address these issues.  I think this may give some insights to the community.

2. The experiments are solid, the authors give many detailed experiments to show their effectiveness, as shown in the manuscript and that in the appendix.

**Weaknesses:**

Actually, I am not very familiar with the backdoor attack and SRV evaluation problem, and just have some background knowledge about them. According to my understanding, there exist some concerns as follows:

1. In my opinion, the trigger in backdoor attack and the real object are different in essence. In an image, the trigger is an out-of-context object, and the real object is in context. Therefore, although a SRV evaluation method can show good performance on the trigger, it does not ensure the effectiveness on the real object. I wonder that the authors whether provide some analysis about this difference.

2. In the experiments, although they conduct many experiments, it seems that there are not some comparisons with the SOTA SRV evaluation methods. In Section 2.2, the authors discuss many existing SRV evaluation methods, they should give the comprehensive comparisons with these methods.

**Questions:**

See the weakness

---

> ### Author Response · Authors · 2023-11-19
> **Author Response**
>
> Dear Reviewer 4nxh, we sincerely thank you for your valuable time and comments. We are encouraged by your positive comments on our
> **reasonable and insightful method**, **solid experiments**, **good contribution**, and **good writing**. We hope the following responses could help clarify potential misunderstandings and alleviate your concerns.
>
>
> ---
> **Q1**: Actually, In my opinion, the trigger in backdoor attack and the real object are different in essence. In an image, the trigger is an out-of-context object, and the real object is in context. Therefore, although a SRV evaluation method can show good performance on the trigger, it does not ensure the effectiveness on the real object. I wonder that the authors whether provide some analysis about this difference.
>
>
> **R1**: Thank you for your insightful question! We admit that our GLBW and other existing backdoor-based SRV methods all rank SRV methods based on their performance in explaining simple backdoored patterns (instead of high-level complicated real object features). However, **it not necessarily means that our method is impractical**. We hereby provide more explanations:
>
> - In practice, **humans categorize a given image based on its local regions in most cases**. For example, we only need to see the image areas of its 'head' to know it is a bird. **These local regions are similar to the trigger patterns** (e.g., a patch) used in our method. Accordingly, results made by our method can be a good reference in real images for practice.
> - Currently, **it is impossible to faithfully evaluate the performance of SRV methods for complicated features** since there is no ground-truth salience map for them. Even a human expert cannot mark the salience map for complicated features.
> - Simple backdoored patterns are also features used by DNNs for their predictions. Accordingly, **the evaluation of SRV methods on trigger features is the first and the most important step toward evaluating their general performance** and is therefore of great significance.
>
> We have provided more details and explanations in Appendix C (*i.e.*, Why Do We (Only) Need to Consider Patch-based Triggers?) of our revision.
>
>
> ---
>
> **Q2**: In the experiments, although they conduct many experiments, it seems that there are not some comparisons with the SOTA SRV evaluation methods. In Section 2.2, the authors discuss many existing SRV evaluation methods, they should give the comprehensive comparisons with these methods.
>
> **R2**: Thank you for this constructive comment! We are deeply sorry that our previous submission may lead you to some misunderstandings that we want to clarify here.
>
> - Currently, the most reliable evaluation of SRV methods is still based on the human inspection. However, **human evaluation is costly and time-consuming**. As such, in practice, we need to design automated assessment methods like our GLBW-based one. Specifically, we have compared human evaluation as a reference to our method in Appendix K to some extent. As shown in the following table 1, **the rankings generated by our GLBW method are similar to those made by people**, verifying the effectiveness of our GLBW.
>
>
> **Table 2.** The average rank of SRV methods that is evaluated with our GLBW method and human inspection on the GTSRB dataset.
> | Dataset$\downarrow$ | Method$\downarrow$, SRV$\rightarrow$ | BP | GBP | GCAM | GGCAM | OCC | FA | LIME |
> |---------|-------------|----|-----|------|-------|-----|----|------|
> | GTSRB   | Ours        | 4  | 4   | 6    | 6     | 2   | 3  | 1    |
> | GTSRB   | Human       | 4  | 4   | 6    | 6     | 2   | 3  | 1    |
>
>
> - For automated evaluation methods, except for the vanilla backdoor-based method compared in our paper, **all other methods have been shown to consume significant computational resources and have limited accuracy** (Hooker el al, 2019). Accordingly, we did not compared our methods with them.
> - Different from classical tasks such as image categorization, **the evaluation of XAI methods is very difficult** because there is no ground-truth saliency areas. However, **backdoor-based methods, especially our GLBW-based one, could lead reliable saliency map evaluation** since the ground-truth saliency areas are those of the trigger pattern **when there is no trigger generalization**. This is why backdoor-based assessment methods are well worth further research.
>
>
> ---

---

> ### Author Response · Authors · 2023-11-21
> **Thanks to Reviewer 4nxh**
>
> Please allow us to thank you again for reviewing our paper and the valuable feedback, and in particular for recognizing the strengths of our paper in terms of *reasonable and insightful method*, *solid experiments*, *good contribution*, and *good writing*.
>
> Kindly let us know if our response and the new experiments have properly addressed your concerns. We are more than happy to answer any additional questions during the post-rebuttal period. Your feedback will be greatly appreciated.

---

> ### Author Response · Authors · 2023-11-22
> **A Gentle Reminder of the Final Feedback**
>
> We would like to thank the reviewer for the helpful discussion during the first round of the review. We hope our response has adequately addressed your concerns. We take this as a great opportunity to improve our work and shall be grateful for any additional feedback you could give to us.

---

> ### Author Response · Authors · 2023-11-23
> **A Second Reminder of the Post-rebuttal Feedback**
>
> Dear Reviewer 4nxh,
>
> We greatly appreciate your initial comments. We totally understand that you may be extremely busy at this time. But we still hope that you could have a quick look at our responses to your concerns. We appreciate any feedback you could give to us. We also hope that you could kindly update the rating if your questions have been addressed. We are also happy to answer any additional questions before the rebuttal ends.
>
> Best Regards,
>
> Paper3310 Authors

---

### Official Review · Reviewer_LeTy · 2023-11-01

**Soundness:** 2 fair
**Presentation:** 2 fair
**Contribution:** 2 fair
**Rating:** 5
**Confidence:** 4

**Summary:**

This paper proposed a generalization-limited backdoor watermark (GLBW), an evaluation method for saliency-based representation visualization (SRV). The author claimed that the existing watermark-based evaluation has a problem with trigger generalization: a typically trained poisoned classifier has a gap between potential (universal adversarial perturbations) and original triggers. Based on this observation, the author proposed GLBW, constructed as a combination of benign, backdoor, and generalization losses to generate generalized samples for training a poisoned model. The author showed that the poisoned model trained with GLBW has better benign accuracy and watermark success rate compared to the baselines. Also, they showed that their standardized version of the backdoor-based saliency map evaluation method has a lower variance than the vanilla method.

**Strengths:**

- This paper addressed a challenging and essential problem: evaluating explanation methods without human annotation.
- They found the implementation problems that exist in the previous watermark-based evaluation method (Lin et al., 2021)

**Weaknesses:**

- Unnecessity of the consideration of trigger generalization in evaluating XAI
  - If I know correctly, the main idea underlying the evaluation of XAI using a backdoor attack is that the poisoned model would refer to the watermarked regions to make a prediction, and a good XAI would highlight these regions as well. In other words, the watermark region is treated as a ground truth of the saliency map. Meanwhile, the author claims that the existence of potential triggers is a problem because it has almost zero cross-entropy loss, but their visualization differs from the original trigger. However, for me, it is unclear why the potential triggers should be considered in evaluating XAI based on watermark regions. For example, assume that there is a cat image and a poisoned model predicts it as a cat. If the prediction results changed by adding a watermark on the image, then it is due to the watermark in a high probability.
- Limitation of IoU based evaluation methods
  - Since the saliency map is not a segmentation, comparing the ground truth segmentation label with the saliency map could not be the best evaluation method. The classifier would not uniformly refer to the ground truth area.
- Week explanation for the intention of BWTP/GLBW
  - Many of the details of BWTP and GLBW are in the appendix, which makes it hard to understand them at first sight.

- Minor corrections
  - In Eq (1), $\mathcal{L}$ receives $y\in${$0,...,K-1$} for benign and backdoor loss, but $\mathbf{y}\in\mathbb{R}^K$ for penalty loss.
  - In Eq (2), $|\mathbf{b}'| is not defined

**Questions:**

- Could you describe how the existence of potential triggers affects the watermark-based saliency map evaluation?
- How do we evaluate which saliency map evaluation metric is better?

---

> ### Author Response · Authors · 2023-11-19
> **Author Response (Part I)**
>
> Dear Reviewer LeTy, we sincerely thank you for your valuable time and comments. We hope the following responses can help clarify potential misunderstandings and alleviate your concerns.
>
> ---
> **Q1**: Unnecessity of the consideration of trigger generalization in evaluating XAI. It is unclear why the potential triggers should be considered in evaluating XAI based on watermark regions. For example, assume that there is a cat image and a poisoned model predicts it as a cat. If the prediction results changed by adding a watermark on the image, then it is due to the watermark in a high probability.
>
>
> **R1**: Thank you for this insightful quesion! We are deeply sorry that we failed to explain it more clearly in our submission. In general, **position generalization will make the result of backdoor-based SRV evaluation less reliable**. More details are as follows.
> - **Backdoor-based SRV evaluation methods rely on a latent assumption that existing backdoor attacks have no trigger generalization**, *i.e.*, only the trigger used for training (dubbed 'original trigger') can activate backdoors. These methods believe that trigger regions should be treated as the regions that contribute the most to the model's prediction (i.e., target label) of poisoned samples because their ground-truth labels are not the target label. Accordingly, they use the area of original trigger as the ground-truth reference and calculate the average intersection over union (IOU) between it and the saliency areas generated by the SRV method of the backdoored model over different backdoored samples as an indicator to evaluate the SRV method.
> - **This assumption does not hold when backdoor watermark has position generalization**. As we shown in our Figure 4, there are many potential trigger patterns other than the original one that can still activate backdoors. In other words, **this assumption does not hold for existing backdoor watermarks**.
> - **Its failure may lead to unreliable results**. For example, given a SRV method, assume that its generated saliency areas of most poisoned samples are only a small part of that of the original trigger. According to the existing backdoor-based evaluation, this SRV method will be treated as having poor performance since it has a small IOU. However, due to the generalization of backdoor watermarks, the model may only learn this local region rather than the whole trigger. In this case, the evaluated SRV method is in fact highly effective, contradicting to the results of the backdoor-based SRV evaluation.
>
> We have added more details in the introduction of our revision to make it more clearly.
>
>
> ---
>
> **Q2**: Limitation of IoU based evaluation methods. Since the saliency map is not a segmentation, comparing the ground truth segmentation label with the saliency map could not be the best evaluation method. The classifier would not uniformly refer to the ground truth area.
>
> **R2**: Thank you for this insightful comment!
> - We admit that classifiers may not uniformly refer to the ground truth area. However, it only leads to the fact that we can hardly obtain the ground-truth saliency areas with their weights. Arguably, **this is not the reason why our IoU-based method is limited**, as all evaluation methods are unable to obtain such ground-truth weights.
> - We never claim that IoU-based evaluation is the best method. We admit that **it may not be the best, but at least it fits** due to the similarities of this task and segmetation.
> - We adopt the IoU-based method **simply following the setting of our baseline (Lin et al., 2021) for a fair comparison**.
> - However, we fully understand your concern about whether using other metrics may change the results. To further alleviate your concerns, we conduct additional experiments on the variant of our standardized method (dubbed 'standardized-L2'), where we adopt $L_2$ distance instead of IoU for the evaluation. As shown in the following table, our standardized-L2 can still reaches consistent rankings. In contrast, the vanilla method has various evaluations of the performance of the same SRV method on different datasets (especially for BP). **These results verify that our evaluation is more faithful and scalable to evaluation metrics**.
>
>
> **Table 1**. The average rank of SRV methods that is evaluated with the vanilla backdoor-based method and its standardized versions on CIFAR-10 and GTSRB datasets.
> | Method$\downarrow$ | Dataset$\downarrow$, SRV$\rightarrow$ | BP | GBP | GCAM | GGCAM | OCC | FA | LIME |
> |:------------:|:--------:|:---:|:---:|:---:|:---:|:---:|:---:|:---:|
> | Vanilla      | CIFAR-10 |5.25 |3 |6.25 |3.25 |1.5 |2.5 |6.25 |
> | Vanilla      | GTSRB    |2.75 |3.75 |6.5 |3.75 |2.5 |2.25 |6.5 |
> | Standardized-IoU | CIFAR-10 |4.75 |4.75 |7 |4.5 |2 |3 |1 |
> | Standardized-IoU | GTSRB    |4.5 |4.5 |7 |5 |2 |2.75 |1.25 |
> | Standardized-L2 | CIFAR-10 |4.5 |4.5 |6.75 |5.25 |2 |3 |1 |
> | Standardized-L2 | GTSRB    |4.75 |4.75 |7 |4.5 |2 |3 |1 |
>
> ---

---

> > ### Author Response · Authors · 2023-11-19
> > **Author Response (Part II)**
> >
> > ---
> >
> > **Q3**: Week explanation for the intention of BWTP/GLBW. Many of the details of BWTP and GLBW are in the appendix, which makes it hard to understand them at first sight.
> >
> > **R3**: Thank you for this constructive comment! Due to space limitations, we only introduced the design ideas and general approaches of our BWTP and GLBW in our submission. We are deeply sorry that we failed to explain them well. We hereby provide more details.
> >
> > - **Backdoor Watermark with Trigger Penalty (BWTP)**. In general, its design is intuitive and straightforward. To make our watermark less generalizable, we need to achieve the two goals: **(1)** If the potential trigger pattern is very close to the ground-truth trigger, then we need to make the prediction closer to the target label. **(2)** If the potential trigger pattern is far from the ground-truth trigger, then we need to make the prediction closer to the ground-truth label. Based on these goals, we designed the **penalty loss to penalize the effects of potential trigger patterns based on its distance to the original one** in each iteration during the training process. To solve the previous optimization problem for model training, we **alternately optimize the inner maximization and the outer minimization in each iteration**.
> >   - **Inner Maximization**: find the worst potential trigger pattern and its mask.
> >   - **Outer Minimization**: update model parameters based on the obtained worst trigger pattern and its mask.
> >
> > - **Generalization-Limited Backdoor Watermark (GLBW)**. However, as shown in our experiments, the performance of **BWTP is unstable regarding both trigger generation and model training**, mostly **due to the complexity of its penalty loss**. To solve this problem, we further simplify the loss term. Inspired by the Neural Cleanse, **we directly identify the potential trigger patterns that are more different from the ground-truth trigger and force them to be predicted as ground-truth labels**, based on which we get our GLBW. Similar to BWTP, we solve GLBW by **alternately optimizing the inner maximization and the outer minimization in each iteration**.
> >
> > We have added more details in Section 4.2-4.3 of our revision to make it more clearly. We are happy to provide more details if you need.
> >
> > ---
> >
> > **Q4**: Minor corrections. In Eq (1), $\mathcal{L}$ receives $y \in 0,...,K-1$ for benign and backdoor loss, but $\mathbf{y}\in\mathbb{R}^K$ for penalty loss. In Eq (2), $|\mathbf{b}'|$ is not defined
> >
> > **R4**: Thank you for pointing them out!
> > - To make it more rigorous, in our revision, we have replaced $\mathcal{L}$ with $\mathcal{L}'$ in the penalty loss and provided more explanations: $\mathcal{L}$ and $\mathcal{L}'$ are the loss functions ($e.g.$, cross-entropy and mutual information).
> > - In fact, there is no $|\mathbf{b}'|$ in our Eq (2). We speculate that you intended to refer to $|\mathbf{m}'|$ instead of $|\mathbf{b}'|$. $|\mathbf{m}'|$ is the size of potential trigger mask $\mathbf{m}'$, which has been defined in Eq (1). To make our Eq (2) more self-consistent, we have added its definition in our revision.
> >
> >
> > ---
> >
> > **Q5**: Could you describe how the existence of potential triggers affects the watermark-based saliency map evaluation?
> >
> > **R5**: Please kindly refer to our responses to **Q1**. We'd be happy to provide more details if you need :)
> >
> > ---
> > **Q6**: How do we evaluate which saliency map evaluation metric is better?
> >
> > **R6**: Thank you for this insightful question!
> >
> > - In general, **our GLBW could lead reliable saliency map evaluation** since the ground-truth saliency areas are those of the trigger pattern when there is no trigger generalization.
> > - We have to admit that our method may not yield the most perfect assessment. This is because our method currently does not completely eliminate trigger generalization, although it has been able to reduce it significantly. **In this case, only human inspection has the best assessment** since this is currently the only way to approximate ground-truth saliency areas. Nonetheless, arguably, our method remains the best automated assessment method and fit for purpose.
> > - To further alleviate your concerns, we have also conducted additional experiments of human inspection in our Appendix K, as shown in the following table. The results show that **the rankings generated by our GLBW method are similar to those made by people**, verifying its effectiveness.
> >
> >
> > **Table 2.** The average rank of SRV methods that is evaluated with our GLBW method and human inspection on the GTSRB dataset.
> > | Dataset$\downarrow$ | Method$\downarrow$, SRV$\rightarrow$ | BP | GBP | GCAM | GGCAM | OCC | FA | LIME |
> > |---------|-------------|----|-----|------|-------|-----|----|------|
> > | GTSRB   | Ours        | 4  | 4   | 6    | 6     | 2   | 3  | 1    |
> > | GTSRB   | Human       | 4  | 4   | 6    | 6     | 2   | 3  | 1    |
> >
> > ---

---

> ### Author Response · Authors · 2023-11-21
> **Thanks to Reviewer LeTy**
>
> Please allow us to thank you again for reviewing our paper and the valuable feedback. These comments are valuable for us to improve our paper further.
>
> Please kindly let us know if our response and the new experiments have properly addressed your concerns. We are more than happy to answer any additional questions during the post-rebuttal period. Your feedback will be greatly appreciated.

---

> ### Author Response · Authors · 2023-11-22
> **A Gentle Reminder of the Final Feedback**
>
> We would like to thank the reviewer for the helpful discussion during the first round of the review. We hope our response has adequately addressed your concerns. We take this as a great opportunity to improve our work and shall be grateful for any additional feedback you could give to us.

---

> ### Author Response · Authors · 2023-11-23
> **A Second Reminder of the Post-rebuttal Feedback**
>
> Dear Reviewer LeTy,
>
> We greatly appreciate your initial comments. We totally understand that you may be extremely busy at this time. But we still hope that you could have a quick look at our responses to your concerns. We appreciate any feedback you could give to us. We also hope that you could kindly update the rating if your questions have been addressed. We are also happy to answer any additional questions before the rebuttal ends.
>
> Best Regards,
>
> Paper3310 Authors

---

> ### Comment · Reviewer_LeTy · 2023-11-23
>
> Thank you for providing a thorough rebuttal to the reviewer's comments. I appreciate the effort you've put into addressing the concerns. Also, I appreciate the use of red text for revised manuscripts.
>
> Many of my concerns are resolved, but I still respectfully disagree with your notions regarding R1. The main argument is whether the existence of potential triggers $x_{adv}$ truly matters for backdoor-based evaluation. To discuss more, I would denote the benign input, poisoned image, and the potential triggers as $x$, $x_p = G(x)$, and $x_{adv}$, respectively.
>
> - Toy example
>
> Consider a specified scenario where the input is denoted as $x\in\mathbb{R}^{d+3}$. In this context, the first $d$ dimensions determine the true label, while the last $3$ dimensions consistently hold zero value according to the original distribution. An adversary trains a poisoned model by manipulating the last three dimensions: the model generates opposite predictions when any of the last three elements is set to 1. Specifically, $G(x; 0) = (x_1, x_2, ..., x_d, 1, 0, 0)$, $G(x; 1) = (x_1, x_2, ..., x_d, 0, 1, 0)$, and $G(x; 2) = (x_1, x_2, ..., x_d, 0, 0, 1)$. For the purpose of backdoor-based XAI evaluation, only $G(x; 0)$ is utilized as a poisoned image. Consequently, $G(x; 1)$ and $G(x; 2)$ remains as potential triggers.
>
> In this case, my opinion is that the existence of potential triggers $G(x; 1)$ and $G(x; 2)$ could not affect the prediction results of  $G(x; 0)$ and so do for the backdoor-based XAI evaluation. The prediction will totally be made by mostly referring to $x_{d+1}$, and the XAI should highlight $x_{d+1}$ as well, regardless of the existence of potential triggers.
>
> - Random location trigger
>
> Consider a poisoned model trained to be activated on the random location trigger, as in Figure 4 (b, d). Obviously, these models will always have potential triggers. The assumption of this paper is that these models cannot be used for a backdoor-based XAI evaluation, regardless of the reliability of this model.
>
> In summary, the main motivation of this paper is from the assumption:
> > Backdoor-based SRV evaluation methods rely on a latent assumption that existing backdoor attacks have no trigger generalization.
>
> I maintain my original stance; I cannot fully endorse the assumption mentioned. This assumption plays a pivotal role throughout the paper, and its validity is of utmost importance. I am inclined to keep my initial score, but I am open to reconsidering and potentially increasing it if the supplement addressing this particular assumption is provided. A more comprehensive discussion or additional supporting evidence on this matter would significantly contribute to strengthening the overall credibility of the paper.

---

> > ### Author Response · Authors · 2023-11-23
> >
> > Thank you for your further insightful comments. We are also glad we addressed all your concerns besides the first one. We hereby provide more details to further alleviate your remaining concern.
> >
> > - We totally agree with you that having trigger generalizability may always influence the predictions and XAI evaluation. However, generalizability does have a significant negative impact in the examples we have mentioned. Arguably, **we can't ignore cases where a method doesn't work just because it works some of the time**. Therefore, reducing generalizability is critical for accurate and faithful assessment.
> > - As for random location triggers, we argue that it will make the evaluation unreliable due to its randomness and generalizability. **This is why our method and the baseline both emphasized the use of a trigger with a fixed location instead of random ones**.
> >
> > Thank you again for your valuable comments and time. They are very important to further improve our work.

---

### Comment · Area_Chair_zAUk · 2023-11-23
**[ICLR 2024 Reviewers’ feedback] Please read authors’ responses and give your feedback**

Dear Reviewers,

Thanks again for your strong support and contribution as an ICLR 2024 reviewer.

Please check the response and other reviewers’ comments. You are encouraged to give authors your feedback after reading their responses. Thanks again for your help!

Best,

AC

---

### Meta-Review · Area_Chair_zAUk · 2023-12-12

**Metareview:**

Most reviewers give high scores. Only one reviewer gave a negative score.

**Justification For Why Not Higher Score:**

Most reviewers give high scores. Only one reviewer gave a negative score.

**Justification For Why Not Lower Score:**

Most reviewers give high scores. Only one reviewer gave a negative score.

---

### Decision · Program_Chairs · 2024-01-16

Accept (poster)